# Error Bounds for Learning with Vector-Valued Random Features

**Samuel Lanthaler**[*]
California Institute of Technology
slanth@caltech.edu

**Nicholas H. Nelsen**[*]
California Institute of Technology
nnelsen@caltech.edu

## Abstract

This paper provides a comprehensive error analysis of learning with vector-valued random features (RF). The theory is developed for RF ridge regression in a fully general infinite-dimensional input-output setting, but nonetheless applies to and improves existing finite-dimensional analyses. In contrast to comparable work in the literature, the approach proposed here relies on a direct analysis of the underlying risk functional and completely avoids the explicit RF ridge regression solution formula in terms of random matrices. This removes the need for concentration results in random matrix theory or their generalizations to random operators. The main results established in this paper include strong consistency of vector-valued RF estimators under model misspecification and minimax optimal convergence rates in the well-specified setting. The parameter complexity (number of random features) and sample complexity (number of labeled data) required to achieve such rates are comparable with Monte Carlo intuition and free from logarithmic factors.

## 1 Introduction

Supervised learning of an unknown mapping $\mathcal{G}\colon \mathcal{X} \to \mathcal{Y}$ is a core task in machine learning. The *random feature model* (RFM), proposed in [36, 37], combines randomization with optimization to accomplish this task. The RFM is based on a linear expansion with respect to a randomized basis, the *random features* (RF). The coefficients in this RF expansion are optimized to fit the given data of input-output pairs. For popular loss functions, such as the square loss, the RFM leads to a convex optimization problem which can be solved efficiently and reliably.

The RFM provides a scalable approximation of an underlying kernel method [36, 37]. While the former is based on an expansion in $M$ random features $\varphi(\,\cdot\,;\theta_1), \ldots, \varphi(\,\cdot\,;\theta_M)$, the corresponding kernel method relies on an expansion in values of a positive definite kernel function $K(\,\cdot\,, u_1), \ldots, K(\,\cdot\,, u_N)$ on a dataset of size $N$. Kernel methods are conceptually appealing, theoretically sound, and have attracted considerable interest [1, 6, 40]. However, they require the storage, manipulation, and often inversion of the kernel matrix $\mathsf{K}$ with entries $K(u_i, u_j)$. The size of $\mathsf{K}$ scales quadratically in the number of samples $N$, which can be prohibitive for large datasets. When the underlying input-output map is vector-valued with $\dim(\mathcal{Y}) = p$, the often significant computational cost of kernel methods is further exacerbated by the fact that each entry $K(u_i, u_j)$ of $\mathsf{K}$ is, in general, a $p$-by-$p$ matrix. Hence, the size of $\mathsf{K}$ scales quadratically in both $N$ and $p$. This severely limits the applicability of kernel methods to problems with high-dimensional, or indeed infinite-dimensional, output space. In contrast, learning with RF only requires storage of RF matrices whose size is quadratic in the number of features $M$. When $M \ll Np$, this implies substantial computational savings, with the most extreme case being the infinite-dimensional setting in which $p = \infty$.

In the context of operator learning, the underlying target mapping is an operator $\mathcal{G}\colon \mathcal{X} \to \mathcal{Y}$ with infinite-dimensional input and output spaces. Such operators appear naturally in scientific computing

---

[*]Equal Contribution

37th Conference on Neural Information Processing Systems (NeurIPS 2023).

and often arise as solution maps of an underlying partial differential equation. Operator learning has attracted considerable interest, e.g., [3, 16, 20, 25, 27], and in this context, the RFM serves as an alternative to neural network-based methods with considerable potential for a sound theoretical basis. Indeed, an extension of the RFM to this infinite-dimensional setting has been proposed and implemented in [32]. Although the results show promise, a mathematical analysis of this approach including error bounds and rates of convergence has so far been outstanding.

**Related work.** Several papers have derived error bounds for learning with RF. Early work on the RFM [37] proceeded by direct inspection of the risk functional, demonstrating that $M \simeq N$ random features suffice to achieve a squared error $O(1/\sqrt{N})$ for RF ridge regression (RR). This result was considerably improved in [38], where $\sqrt{N} \log N$ random features were shown to be sufficient to achieve the same squared error. This improvement in parameter complexity is based on the explicit RF RR solution formula, combined with extensive use of matrix analysis and matrix concentration inequalities. Similar analysis in [26] sharpens the parameter complexity to $\sqrt{N} \log d_{\mathsf{K}}^\lambda$ random features. Here $d_{\mathsf{K}}^\lambda$ is the *number of effective degrees of freedom* [2, 6], with $\lambda$ the RR regularization parameter and $\mathsf{K}$ the kernel matrix. In this context, we also mention related analysis in [2]. In all these works, the squared error in terms of sample size $N$ match the minimax optimal rates for kernel RR derived in [6]. Going beyond the above error bounds, [2, 26, 38] also derive fast rates under additional assumptions on the underlying data distribution and/or with improved RF sampling schemes.

Many works also study the interpolatory ($M \simeq N$) or overparametrized ($M \gg N$) regimes in the scalar output setting [10, 14, 15, 18, 29]. However, when $p = \dim(\mathcal{Y}) \gg 1$ or $p = \infty$, such regimes may no longer be relevant. This is because the kernel matrix $\mathsf{K}$ now has size $Np$-by-$Np$, and it is possible that the number of random features $M$ satisfies $M \ll Np$ even though $M \gg N$. In this case, high-dimensional vector-valued learning naturally operates in the underparametrized regime.

In the area of operator learning for scientific problems, approximation results are common [11, 13, 19, 17, 22, 24, 39, 41] but statistical guarantees are lacking, the main exceptions being [4, 12, 21, 31, 43] in the linear operator setting and [6] in the nonlinear setting. The RFM also has potential for such nonlinear problems. Indeed, vector-valued random Fourier features have been studied before [5, 30]. However, theory is only provided for kernel approximation, not generalization guarantees.

To summarize, while previous analyses have provided considerable insight into the generalization properties of the RFM, they have almost exclusively focused on the scalar-valued setting. Given the paucity of theoretical work beyond this setting, it is a priori unclear whether similar estimates continue to hold when the RFM is applied to infinite-dimensional vector-valued mappings.

**Contributions.** The primary purpose of this paper is to extend earlier results on learning with random features to the vector-valued setting. The theory developed in this work unifies sources of error stemming from approximation, generalization, misspecification, and noisy observations. We focus on training via ridge regression with the square loss. Our results differ from existing work not only in the scope of applicability, but also in the strategy employed to derive our results. Similar to [37], we do not rely on the explicit random feature ridge regression solution (RF-RR) formula, which is specific to the square loss. One main benefit of this approach is that it entirely avoids the use of matrix concentration inequalities, thereby making the extension to an infinite-dimensional vector-valued setting straightforward. Our main contributions are now listed (see also Table 1).

(C1) Given $N$ training samples, we prove that $M \simeq \sqrt{N}$ random features and regularization strength $\lambda \simeq 1/\sqrt{N}$ is enough to guarantee that the squared error is $O(1/\sqrt{N})$, provided that the target operator belongs to a specific reproducing kernel Hilbert space (Thm. 3.7);

(C2) we establish that the vector-valued RF-RR estimator is strongly consistent (Thm. 3.10);

(C3) under additional regularity assumptions, we derive rates of convergence even when the target operator does not belong to the specific reproducing kernel Hilbert space (Thm. 3.12);

(C4) we demonstrate that the approach of Rahimi and Recht [37] can be used to derive state-of-the-art rates for the RFM which, for the first time, are free from logarithmic factors.

**Outline.** The remainder of this paper is organized as follows. We set up the ridge regression problem in Sect. 2. The main results are stated in Sect. 3 and their proofs are sketched in Sect. 4.

Sect. 5 provides a simulation study and Sect. 6 gives concluding remarks. Detailed proofs are deferred to the supplementary material (**SM**).

Table 1: A summary of available error estimates for the RFM, with regularization parameter $\lambda$, output space $\mathcal{Y}$, and number of random features $M$. $(^\star)$: the truth is assumed to be written as $\mathcal{G}(u) = \mathbb{E}_\theta[\alpha^*(\theta)\varphi(u;\theta)]$ with restrictive bound $|\alpha^*(\theta)| \leq R$ to avoid explicit regularization.

| Paper | Approach | $\lambda$ | $\dim(\mathcal{Y})$ | $M$ | Squared Error |
|---|---|---|---|---|---|
| Rahimi & Recht [37] | "kitchen sinks" | n/a $(^\star)$ | 1 | $N$ | $R/\sqrt{N}$ |
| Rudi & Rosasco [38] | matrix concen. | $1/\sqrt{N}$ | 1 | $\sqrt{N}\log(N)$ | $1/\sqrt{N}$ |
| Li et al. [26] | matrix concen. | $1/\sqrt{N}$ | 1 | $\sqrt{N}\log(d_{\mathsf{K}}^\lambda)$ | $1/\sqrt{N}$ |
| **This work** | **"kitchen sinks"** | $\mathbf{1/\sqrt{N}}$ | $\boldsymbol{\infty}$ | $\boldsymbol{\sqrt{N}}$ | $\mathbf{1/\sqrt{N}}$ |

## 2 Preliminaries

We now set up our vector-valued learning framework by introducing notational conventions, reviewing random features, and formulating the ridge regression problem.

**Notation.** Let $(\Omega, \mathcal{F}, \mathbb{P})$ be a sufficiently rich probability space on which all random variables in this paper are defined. Let $\mathcal{X}$ be the input space, $\mathcal{Y}$ the output space, and $\Theta$ a set. We consistently use $u$ to denote elements of $\mathcal{X}$ and $\theta$ for RF parameters in $\Theta$. The set of probability measures supported on a set $\mathcal{Q}$ is denoted by $\mathscr{P}(\mathcal{Q})$. We write expectation (in the sense of Bochner integration) with respect to $u \sim \nu \in \mathscr{P}(\mathcal{X})$ as $\mathbb{E}_u[\,\cdot\,]$ and similarly for $\theta \sim \mu \in \mathscr{P}(\Theta)$. Independent and identically distributed (iid) samples $u_1, \ldots, u_N$ from $\nu$ will be denoted by $\{u_n\} \sim \nu^{\otimes N}$ and similarly for $\{\theta_m\} \sim \mu^{\otimes M}$. We write $a \simeq b$ to mean that there exists a constant $C \geq 1$ such that $C^{-1}b \leq a \leq Cb$ and similarly for the one-sided inequalities $a \lesssim b$ and $a \gtrsim b$. We define $a \wedge b := \min(a, b)$.

**Random features and reproducing kernel Hilbert spaces.** Random features are defined by a pair $(\varphi, \mu)$, where $\varphi \colon \mathcal{X} \times \Theta \to \mathcal{Y}$ and $\mu \in \mathscr{P}(\Theta)$. Fixing $\theta \sim \mu$ defines a map $\varphi(\,\cdot\,;\theta) \colon \mathcal{X} \to \mathcal{Y}$. Considering linear combinations of such maps leads to the following definition.

**Definition 2.1** (Random feature model). The map $\Phi(\,\cdot\,;\alpha) = \Phi(\,\cdot\,;\alpha, \{\theta_m\}) \colon \mathcal{X} \to \mathcal{Y}$ given by

$$u \mapsto \Phi(u;\alpha) := \frac{1}{M}\sum_{m=1}^{M}\alpha_m\varphi(u;\theta_m) \tag{2.1}$$

is a *random feature model* (RFM) with coefficients $\alpha \in \mathbb{R}^M$ and fixed realizations $\{\theta_m\} \sim \mu^{\otimes M}$.

Associated to the pair $(\varphi, \mu)$ is a reproducing kernel Hilbert space (RKHS) $\mathcal{H}$ of maps from $\mathcal{X}$ to $\mathcal{Y}$ [32, Sect. 2.3]. Under mild assumptions (see **SM B**) assumed in our main results, it holds that

$$\mathcal{H} = \left\{ \mathcal{G} \in L^2_\nu(\mathcal{X}; \mathcal{Y}) \,\middle|\, \mathcal{G} = \mathbb{E}[\alpha(\theta)\varphi(\,\cdot\,;\theta)] \text{ and } \alpha \in L^2_\mu(\Theta; \mathbb{R}) \right\} \tag{2.2}$$

with RKHS norm $\|\mathcal{G}\|_{\mathcal{H}} = \min_\alpha \|\alpha\|_{L^2_\mu}$, where $\alpha$ ranges over all decompositions of $\mathcal{G}$ of the form in (2.2). A minimizer $\alpha_{\mathcal{H}}$ of this problem always exists [2, Sect. 2.2]. We use this fact to identify any $\mathcal{G} \in \mathcal{H}$ with its minimizing $\alpha_{\mathcal{H}} \in L^2_\mu(\Theta; \mathbb{R})$ without further comment.

**Random feature ridge regression.** Let $\mathcal{P} \in \mathscr{P}(\mathcal{X} \times \mathcal{Y})$ be the *joint data distribution*. The goal of RF-RR is to estimate an underlying operator $\mathcal{G} \colon \mathcal{X} \to \mathcal{Y}$ from finitely many iid input-output pairs $\{(u_n, y_n)\}_{n=1}^N \sim \mathcal{P}^{\otimes N}$, where typically the $y_n$ are noisy transformations of the point values $\mathcal{G}(u_n)$. To describe RF-RR, we first make some definitions.

**Definition 2.2** (Empirical risk). Writing $Y = \{y_n\}$ for the collection of observed output data and fixing a regularization parameter $\lambda > 0$, the *regularized $Y$-empirical risk* of $\alpha \in \mathbb{R}^M$ is given by

$$\mathscr{R}_N^\lambda(\alpha; Y) := \frac{1}{N}\sum_{n=1}^N \|y_n - \Phi(u_n;\alpha)\|_{\mathcal{Y}}^2 + \lambda\|\alpha\|_M^2, \quad \text{where} \quad \|\alpha\|_M^2 := \frac{1}{M}\sum_{m=1}^M |\alpha_m|^2 \tag{2.3}$$

is a scaled Euclidean norm on $\mathbb{R}^M$. The *regularized $\mathcal{G}$-empirical risk*, $\mathscr{R}_N^\lambda(\alpha; \mathcal{G})$, is defined analogously with $\mathcal{G}(u_n)$ in place of $y_n$. In the absence of regularization, i.e., $\lambda = 0$, these expressions define the *$Y$-empirical risk* and *$\mathcal{G}$-empirical risk*, denoted by $\mathscr{R}_N(\alpha; Y)$ and $\mathscr{R}_N(\alpha; \mathcal{G})$, respectively.

RF-RR is the minimization problem $\min_{\alpha \in \mathbb{R}^M} \mathscr{R}_N^\lambda(\alpha; Y)$. The minimizer, which we denote by $\widehat{\alpha}$, is referred to as *trained coefficients* and $\Phi(\,\cdot\,; \widehat{\alpha})$ the *trained RFM*. For $M$ and $N$ sufficiently large and $\lambda > 0$ sufficiently small, we expect the trained RFM to well approximate $\mathcal{G}$. This intuition is made precise by quantitative error bounds and statistical performance guarantees in the next section.

## 3 Main results

The main result of this paper is an abstract bound on the population squared error (Sect. 3.2). From this widely applicable theorem, several more specialized results are deduced. These include consistency (Sect. 3.3) and convergence rates (Sect. 3.4) of the RF-RR estimator trained on noisy data. The assumptions under which this theory is developed are provided next in Sect. 3.1.

### 3.1 Assumptions

Throughout this paper, we assume that the input space $\mathcal{X}$ is a Polish space and the output space $\mathcal{Y}$ is a real separable Hilbert space. These are common assumptions in learning theory [6]. We view $\mathcal{X}$ and $\mathcal{Y}$ as measurable spaces equipped with their respective Borel $\sigma$-algebras.

Next, we make the following minimal assumptions on the random feature pair $(\varphi, \mu)$.

**Assumption 3.1** (Random feature regularity). *Let $\nu \in \mathscr{P}(\mathcal{X})$ be the input distribution and $(\Theta, \Sigma, \mu)$ be a probability space. The random feature map $\varphi \colon \mathcal{X} \times \Theta \to \mathcal{Y}$ and the probability measure $\mu \in \mathscr{P}(\Theta)$ are such that (i) $\varphi$ is measurable; (ii) $\varphi$ is uniformly bounded; in fact, $\|\varphi\|_{L^\infty} :=$ $\operatorname{ess\,sup}_{(u,\theta) \sim \nu \otimes \mu} \|\varphi(u; \theta)\|_{\mathcal{Y}} \leq 1$; and (iii) the RKHS $\mathcal{H}$ corresponding to $(\varphi, \mu)$ is separable.*

The boundedness assumption on $\varphi$ is shared in general theoretical analyses of RF [26, 37, 38]; the unit bound can always be ensured by a simple rescaling. We work in a general misspecified setting.

**Assumption 3.2** (Misspecification). *There exist $\rho \in L_\nu^\infty(\mathcal{X}; \mathcal{Y})$ and $\mathcal{G}_\mathcal{H} \in \mathcal{H}$ such that the operator $\mathcal{G} \colon \mathcal{X} \to \mathcal{Y}$ satisfies the decomposition $\mathcal{G} = \rho + \mathcal{G}_\mathcal{H}$.*

Since Assumption 3.1 implies that $\mathcal{H} \subset L_\nu^\infty(\mathcal{X}; \mathcal{Y})$, any $\mathcal{G} = \mathcal{G}_\mathcal{H} + \rho$ as in Assumption 3.2 is automatically bounded in the sense that $\mathcal{G} \in L_\nu^\infty(\mathcal{X}; \mathcal{Y})$. Conversely, any $\mathcal{G} \in L_\nu^\infty(\mathcal{X}; \mathcal{Y})$ allows such a decomposition. We interpret $\rho$ as a residual from the operator $\mathcal{G}_\mathcal{H}$ belonging to the RKHS. It may be prescribed by the problem, as we will see later in the context of discretization errors in operator learning (Ex. 3.9), or be arbitrary, as is customary in learning theory when the only information about $\mathcal{G}$ is that it is bounded.

Our main goal is to recover $\mathcal{G}$ from iid data $\{(u_n, y_n)\}$ arising from the following statistical model.

**Assumption 3.3** (Joint data distribution). *The joint distribution $\mathcal{P} \in \mathscr{P}(\mathcal{X} \times \mathcal{Y})$ of the random variable $(u, y) \sim \mathcal{P}$ is given by $u \sim \nu$ with $\nu \in \mathscr{P}(\mathcal{X})$ and $y = \mathcal{G}(u) + \eta$. Here, $\mathcal{G}$ satisfies Assumption 3.2. The additive noise $\eta$ is a random variable in $\mathcal{Y}$ that is conditionally centered, $\mathbb{E}[\eta \,|\, u] = 0$, and is subexponential: $\|\eta\|_{\psi_1(\mathcal{Y})} < \infty$; see (A.7) for the definition of $\|\cdot\|_{\psi_1(\mathcal{Y})}$.*

Assumption 3.3 implies that $\mathcal{G}(u) = \mathbb{E}[y \,|\, u]$. In contrast to related work [2, 26, 37], we allow for unbounded input-dependent noise. In particular, our results also hold for bounded or subgaussian noise, as well as *multiplicative noise* (e.g., $\eta = \xi \mathcal{G}(u)$ with $\mathbb{E}[\xi \,|\, u] = 0$ and $\|\xi\|_{\psi_1} < \infty$).

### 3.2 General error bound

For any $\mathcal{G}$, define the $\mathcal{G}$-*population risk functional* or $\mathcal{G}$-*population squared error* by

$$\mathscr{R}(\alpha; \mathcal{G}) := \mathbb{E}_{u \sim \nu} \|\mathcal{G}(u) - \Phi(u; \alpha, \{\theta_m\})\|_{\mathcal{Y}}^2 \quad \text{for} \quad \alpha \in \mathbb{R}^M . \tag{3.1}$$

The main result of this paper establishes an upper bound for this quantity that holds with high probability, provided that the number of random features and number of data pairs are large enough.

**Theorem 3.4** ($\mathcal{G}$-population squared error bound). *Suppose that $\mathcal{G} = \rho + \mathcal{G}_\mathcal{H}$ satisfies Assumption 3.2. Fix a failure probability $\delta \in (0, 1)$, regularization strength $\lambda \in (0, 1)$, and sample size $N$. Let $\{\theta_m\} \sim \mu^{\otimes M}$ be the $M$ random feature parameters and $\{(u_n, y_n)\} \sim \mathcal{P}^{\otimes N}$ be the data according to Assumption 3.3. For $\Phi$ the RFM (2.1) satisfying Assumption 3.1, let $\widehat{\alpha} \in \mathbb{R}^M$ be the minimizer*

of the regularized $Y$-empirical risk $\mathscr{R}_N^\lambda(\,\cdot\,;Y)$ given by (2.3). If $M \geq \lambda^{-1}\log(32/\delta)$ and $N \geq \lambda^{-2}\log(16/\delta)$, then

$$\mathbb{E}_{u\sim\nu}\|\mathcal{G}(u) - \Phi(u;\widehat{\alpha},\{\theta_m\})\|_{\mathcal{Y}}^2 \leq 79e^{3/2}\big(\|\mathcal{G}\|_{L_\nu^\infty}^2 + 2\beta(\rho,\lambda,\mathcal{G}_\mathcal{H},\eta)\big)\lambda \tag{3.2}$$

with probability at least $1 - \delta$, where

$$\beta(\rho,\lambda,\mathcal{G}_\mathcal{H},\eta) \coloneqq 328\|\mathcal{G}_\mathcal{H}\|_\mathcal{H}^2 + 2023e^3\|\eta\|_{\psi_1(\mathcal{Y})}^2 + 8\lambda^{-1}\mathbb{E}_{u\sim\nu}\|\rho(u)\|_{\mathcal{Y}}^2 + 18\lambda\|\rho\|_{L_\nu^\infty}^2 \tag{3.3}$$

is a function of $\rho$, $\lambda$, $\mathcal{G}_\mathcal{H}$, and the law of the noise variable $\eta$.

The main elements of the proof of Thm. 3.4 will be explained in Sect. 4.

***Remark*** 3.5 (Excess risk). We note that other work [26, 37, 38] often focuses on bounding the *excess risk* $\widehat{\mathscr{E}} \coloneqq \mathscr{E}(\Phi(\,\cdot\,;\widehat{\alpha})) - \inf_{\mathcal{G}_\mathcal{H}\in\mathcal{H}}\mathscr{E}(\mathcal{G}_\mathcal{H})$, where $\mathscr{E}(F) \coloneqq \mathbb{E}\|y - F(u)\|_{\mathcal{Y}}^2 = \mathbb{E}_{u\sim\nu}\|\mathcal{G}(u) - F(u)\|_{\mathcal{Y}}^2 + \mathbb{E}\|\eta\|_{\mathcal{Y}}^2$. In particular, this bias-variance decomposition implies that $\widehat{\mathscr{E}} \leq \mathbb{E}_{u\sim\nu}\|\mathcal{G}(u) - \Phi(u;\widehat{\alpha})\|_{\mathcal{Y}}^2$. Thus, Thm. 3.4 also gives a corresponding bound on the excess risk $\widehat{\mathscr{E}}$.

***Remark*** 3.6 (The $\beta$ factor). In the well-specified setting, that is, $\mathcal{G} - \mathcal{G}_\mathcal{H} = \rho \equiv 0$, the factor $\beta$ in Thm. (3.4) satisfies the uniform bound

$$\beta(\rho,\lambda,\mathcal{G}_\mathcal{H},\eta) \leq B \coloneqq 328\|\mathcal{G}\|_\mathcal{H}^2 + 2023e^3\|\eta\|_{\psi_1(\mathcal{Y})}^2. \tag{3.4}$$

In particular, the constant $B$ does not depend on $\lambda$ in this case. Otherwise, $\beta$ in general depends on $\lambda$. We can characterize this dependence precisely if it is known that $\mathcal{G} \in L_\nu^\infty(\mathcal{X};\mathcal{Y})$. In this case, Assumption 3.2 is satisfied with $\rho \coloneqq \mathcal{G} - \mathcal{G}_\mathcal{H}$ for any $\mathcal{G}_\mathcal{H} \in \mathcal{H}$. Choosing $\mathcal{G}_\mathcal{H} = \mathcal{G}_\vartheta|_{\vartheta=\lambda}$ as in **SM** B (which is optimal in the sense described there) and a short calculation deliver the bound

$$\beta(\rho,\lambda,\mathcal{G}_\mathcal{H},\eta) \lesssim \lambda^{-1}\lambda^{r\wedge1} = \lambda^{-(1-r)_+} \tag{3.5}$$

if $\mathcal{G}$ additionally satisfies a particular $r$-th order regularity condition (see Lem. B.3 for the details). Here, $a_+ \coloneqq \max(a,0)$ for any $a \in \mathbb{R}$. Thus, $\beta$ is uniformly bounded if $\mathcal{G} \in \mathcal{H}$ ($r \geq 1$) and grows algebraically as a power of $\lambda^{-1}$ otherwise ($0 \leq r < 1$).

**Consequences.** The general error bound (3.2) in Thm. 3.4 has several implications for vector-valued learning with the RFM. First, it immediately implies a rate of convergence if $\mathcal{G} \in \mathcal{H}$.

**Theorem 3.7** (Well-specified). *Instantiate the hypotheses and notation of Thm. 3.4. Suppose that $\rho \equiv 0$ so that $\mathcal{G} \in \mathcal{H}$ (2.2). If $M \geq \lambda^{-1}\log(32/\delta)$ and $N \geq \lambda^{-2}\log(16/\delta)$, then*

$$\mathbb{E}_{u\sim\nu}\|\mathcal{G}(u) - \Phi(u;\widehat{\alpha})\|_{\mathcal{Y}}^2 \leq 79e^{3/2}\big(\|\mathcal{G}\|_{L_\nu^\infty}^2 + 2B\big)\lambda \lesssim \lambda \tag{3.6}$$

*with probability at least $1 - \delta$, where the constant $B \geq 0$ is defined by (3.4).*

Given a number of samples $N$, Thm. 3.7 shows that RF-RR with regularization $\lambda \simeq 1/\sqrt{N}$ and number of features $M \gtrsim \sqrt{N}$ leads to a population squared error of size $1/\sqrt{N}$ with high probability. This result should be compared to the previous state-of-the-art convergence rates in the literature for RF-RR with iid sampled features [2, 26, 37, 38]. See Table 1, which indicates that our analysis gives the lowest parameter complexity to date. We emphasize that such a convergence rate rests on the assumption that $\mathcal{G} \in \mathcal{H}$. This corresponds to a *compatibility condition* between $\mathcal{G}$ and the pair $(\varphi,\mu)$, i.e., the random feature map $\varphi$ and the probability measure $\mu$, which determine the RKHS $\mathcal{H}$. Designing suitable $\varphi$ and $\mu$ for a given operator $\mathcal{G}$ remains an open problem. For an explanation of the poor parameter complexity in Rahimi and Recht's original paper [37], see [44, Appendix E].

Thm. 3.4 also implies convergence of $\mathscr{R}(\widehat{\alpha};\mathcal{G})$ when $\mathcal{G} \notin \mathcal{H}$, as we will see in Sect. 3.3 and 3.4. But first, the next corollary shows that the same general bound (3.2) also holds for the $\mathcal{G}_\mathcal{H}$-population squared error $\mathscr{R}(\widehat{\alpha};\mathcal{G}_\mathcal{H})$, up to enlarged constant factors. The proof is given in **SM** C.

**Corollary 3.8** ($\mathcal{G}_\mathcal{H}$-population squared error bound). *Instantiate the hypotheses and notation of Thm. 3.4. If $M \geq \lambda^{-1}\log(32/\delta)$ and $N \geq \lambda^{-2}\log(16/\delta)$, then there exists an absolute constant $C > 1$ such that with probability at least $1 - \delta$, it holds that*

$$\mathbb{E}_{u\sim\nu}\|\mathcal{G}_\mathcal{H}(u) - \Phi(u;\widehat{\alpha})\|_{\mathcal{Y}}^2 \leq C\big(\|\mathcal{G}\|_{L_\nu^\infty}^2 + 2\beta(\rho,\lambda,\mathcal{G}_\mathcal{H},\eta)\big)\lambda. \tag{3.7}$$

Although our main goal is to learn $\mathcal{G}$ from noisy data, there are settings instead in which the learning of $\mathcal{G}_\mathcal{H} \in \mathcal{H}$ as in Cor. 3.8 is of primary interest, but only values of some approximation $\mathcal{G} \in L_\nu^\infty(\mathcal{X};\mathcal{Y})$ are available. The following example illustrates this.

***Example*** **3.9** (Numerical discretization error). One practically relevant setting to which Cor. 3.8 applies arises when training a RFM from functional data generated by a numerical approximation $\mathcal{G} = \mathcal{G}_\Delta$ of some underlying operator $\mathcal{G}_\mathcal{H} \in \mathcal{H}$. Here $\Delta > 0$ represents a numerical parameter, such as the grid resolution when approximating the solution operator of a partial differential equation. In this setting, $\rho = \mathcal{G}_\Delta - \mathcal{G}_\mathcal{H}$ is non-zero and it is crucial to include the discretization error in the analysis, which we define as $\varepsilon_\Delta := \|\rho\|^2_{L^\infty_\nu} = \|\mathcal{G}_\Delta - \mathcal{G}_\mathcal{H}\|^2_{L^\infty_\nu}$. Assume $\eta \equiv 0$, so that $\widehat{\alpha}$ minimizes $\mathscr{R}^\lambda_N(\,\cdot\,;Y) = \mathscr{R}^\lambda_N(\,\cdot\,;\mathcal{G}_\Delta)$. Using Cor. 3.8, it follows that for $M$ and $N$ sufficiently large,

$$\mathbb{E}_{u\sim\nu}\|\mathcal{G}_\mathcal{H}(u) - \Phi(u;\widehat{\alpha})\|^2_\mathcal{Y} \lesssim \lambda\|\mathcal{G}_\mathcal{H}\|^2_\mathcal{H} + \varepsilon_\Delta \tag{3.8}$$

with high probability. Thus, as suggested by intuition, in addition to the error contribution that is present when training on perfect data (the first term on the right-hand side), there is an additional discretization error of size $\varepsilon_\Delta$. We also see that the performance of RF-RR is stable with respect to such discretization errors stemming from the training data. Actually obtaining a rate of convergence would require problem-specific information about the particular numerical solver that is used.

## 3.3 Statistical consistency

We now return to the objective of recovering $\mathcal{G}$ from data. In particular, suppose that $\mathcal{G} \notin \mathcal{H}$; the RKHS, viewed as a hypothesis class, is misspecified. Our analysis demonstrates that statistical guarantees for RF-RR are still possible in this setting.

To this end, assume that $\mathcal{G} \in L^\infty_\nu(\mathcal{X};\mathcal{Y})$. It follows that Assumption 3.2 is satisfied with $\rho := \mathcal{G} - \mathcal{G}_\mathcal{H}$ and $\mathcal{G}_\mathcal{H} \in \mathcal{H}$ being *any* element of the RKHS. Applying Thm. 3.4 and minimizing over $\mathcal{G}_\mathcal{H}$ yields

$$\mathbb{E}_{u\sim\nu}\|\mathcal{G}(u) - \Phi(u;\widehat{\alpha})\|^2_\mathcal{Y} \lesssim \lambda + \inf_{\mathcal{G}_\mathcal{H}\in\mathcal{H}}\left\{\mathbb{E}_{u\sim\nu}\|\mathcal{G}(u) - \mathcal{G}_\mathcal{H}(u)\|^2_\mathcal{Y} + \lambda\|\mathcal{G}_\mathcal{H}\|^2_\mathcal{H}\right\} \tag{3.9}$$

with probability at least $1 - \delta$ if $M \gtrsim \lambda^{-1}\log(2/\delta)$ and $N \gtrsim \lambda^{-2}\log(2/\delta)$. To obtain (3.9), we enlarged constants and used the bound $\|\rho\|^2_{L^\infty_\nu} \lesssim \|\mathcal{G}\|^2_{L^\infty_\nu} + \|\mathcal{G}_\mathcal{H}\|^2_{L^\infty_\nu}$ in (3.3).

If $\mathcal{G}$ is in the $L^2_\nu$-closure of $\mathcal{H}$, then with high probability, the population squared error on the left hand side of (3.9) converges to zero as $\lambda \to 0$ (by application of Lem. B.2 to the second term on the right). This is a statement about the (weak) *statistical consistency* of the trained RF-RR estimator; it can be upgraded to an almost sure statement, as expressed precisely in the next main result.

**Theorem 3.10** (Strong consistency). *Suppose that $\mathcal{G} \in L^\infty_\nu(\mathcal{X};\mathcal{Y})$ belongs to the $L^2_\nu(\mathcal{X};\mathcal{Y})$-closure of $\mathcal{H}$ (2.2). Let $\{\lambda_k\}_{k\in\mathbb{N}} \subset (0,1)$ be a sequence of positive regularization parameters such that $\sum_{k\in\mathbb{N}} \lambda_k < \infty$. For $\Phi$ the RFM (2.1) satisfying Assumption 3.1 and for each $k$, let $\widehat{\alpha}^{(k)} \in \mathbb{R}^{M_k}$ be the trained RFM coefficients that minimize the regularized $Y$-empirical risk $\mathscr{R}^{\lambda_k}_{N_k}(\,\cdot\,;Y)$ given by (2.3) with $M_k \simeq \lambda_k^{-1}\log(2/\lambda_k)$ iid random features and $N_k \simeq \lambda_k^{-2}\log(2/\lambda_k)$ iid data pairs under Assumption 3.3. It holds true that*

$$\lim_{k\to\infty} \mathbb{E}_{u\sim\nu}\|\mathcal{G}(u) - \Phi(u;\widehat{\alpha}^{(k)})\|^2_\mathcal{Y} = 0 \;\; \text{with probability one.} \tag{3.10}$$

The proof relies on a standard Borel–Cantelli argument. See **SM C** for the details.

***Remark*** **3.11** (Universal RKHS). The assumption that $\mathcal{G}$ belongs to the $L^2_\nu$-closure of the RKHS $\mathcal{H}$ is automatically satisfied if $\mathcal{H}$ is dense in $L^2_\nu(\mathcal{X};\mathcal{Y})$. This is equivalent to its kernel being *universal* [7, 9]. In this case, the trained RFM is a strongly consistent estimator of any $\mathcal{G} \in L^\infty_\nu$. However, we are unaware of any practical characterizations of universality of the kernel in terms of its corresponding random feature pair $(\varphi, \mu)$ for the vector-valued setting studied here.

## 3.4 Convergence rates

The previous subsection establishes convergence guarantees without any rates. We now establish quantitative bounds. Throughout what follows, we denote by $\mathcal{K}\colon L^2_\nu(\mathcal{X};\mathcal{Y}) \to L^2_\nu(\mathcal{X};\mathcal{Y})$ the integral operator (B.5) corresponding to the operator-valued kernel function of the RKHS $\mathcal{H}$ (see **SM B**).

**Theorem 3.12** (Slow rates under misspecification). *Suppose that $\mathcal{G} \in L^\infty_\nu(\mathcal{X};\mathcal{Y})$ and that Assumption 3.3 holds. Additionally, assume that $\mathcal{G} \in \text{Im}(\mathcal{K}^{r/2})$ for some $r > 0$, where $\mathcal{K}$ is the integral operator corresponding to the kernel of RKHS $\mathcal{H}$ (2.2). Fix $\delta \in (0,1)$ and $\lambda \in (0,1)$. For*

$\Phi$ *the RFM* (2.1) *satisfying Assumption* 3.1, *let* $\widehat{\alpha} \in \mathbb{R}^M$ *minimize* $\mathscr{R}_N^\lambda(\,\cdot\,; Y)$ *given by* (2.3). *If* $M \geq \lambda^{-1} \log(32/\delta)$ *and* $N \geq \lambda^{-2} \log(16/\delta)$, *then with probability at least* $1 - \delta$ *it holds that*

$$\mathbb{E}_{u \sim \nu} \|\mathcal{G}(u) - \Phi(u; \widehat{\alpha})\|_{\mathcal{Y}}^2 \lesssim \lambda^{r \wedge 1} \,. \tag{3.11}$$

*The implied constant in* (3.11) *depends only on* $\|\mathcal{G}\|_{L_\nu^\infty}$ *and* $\|\eta\|_{\psi_1(\mathcal{Y})}$.

Thm. 3.12 provides a quantitative convergence rate as $\lambda \to 0$. For $r \geq 1$, i.e., when $\mathcal{G} \in \mathcal{H}$, we recover the linear convergence rate of order $\lambda$ from Thm. 3.7. The assumption that $\mathcal{G} \in \mathrm{Im}(\mathcal{K}^{r/2})$ can be viewed as a "fractional regularity" assumption on the underlying operator; indeed, in specific settings it corresponds to a fractional (Sobolev) regularity of the underlying function. In general, it appears difficult to check this condition in practice, which is one limitation of our result.

A quantitative analog to the almost sure statement of Thm. 3.10 also holds.

**Corollary 3.13** (Strong convergence rate)**.** *Instantiate the hypotheses and notation of Thm.* 3.10. *Assume in addition that* $\mathcal{G} \in \mathrm{Im}(\mathcal{K}^{r/2})$ *for some* $r > 0$. *Let* $\{\lambda_k\}_{k \in \mathbb{N}} \subset (0, 1)$ *be a sequence of positive regularization parameters such that* $\sum_{k \in \mathbb{N}} \lambda_k < \infty$. *For each* $k$, *let* $\widehat{\alpha}^{(k)} \in \mathbb{R}^{M_k}$ *be the trained RFM coefficients with* $M_k \simeq \lambda_k^{-1} \log(2/\lambda_k)$ *and* $N_k \simeq \lambda_k^{-2} \log(2/\lambda_k)$. *It holds true that*

$$\limsup_{k \to \infty} \left( \frac{\mathbb{E}_{u \sim \nu} \|\mathcal{G}(u) - \Phi(u; \widehat{\alpha}^{(k)})\|_{\mathcal{Y}}^2}{\lambda_k^{r \wedge 1}} \right) < \infty \quad \text{with probability one.} \tag{3.12}$$

Short proofs of both Thm. 3.12 and Cor. 3.13 may be found in **SM C**.

# 4 Proof outline for the main theorem

Our main results are all derived from Thm. 3.4, whose proof, schematically illustrated in Figure 1, we now outline. Following [37], we break the proof into several steps that arise from the error decomposition

$$\mathscr{R}(\widehat{\alpha}; \mathcal{G}) = \mathscr{R}_N(\widehat{\alpha}; \mathcal{G}) + \left[ \mathscr{R}(\widehat{\alpha}; \mathcal{G}) - \mathscr{R}_N(\widehat{\alpha}; \mathcal{G}) \right] \,. \tag{4.1}$$

Sect. 4.1 estimates the first term on the right hand side of (4.1) while Sect. 4.2 estimates the second.

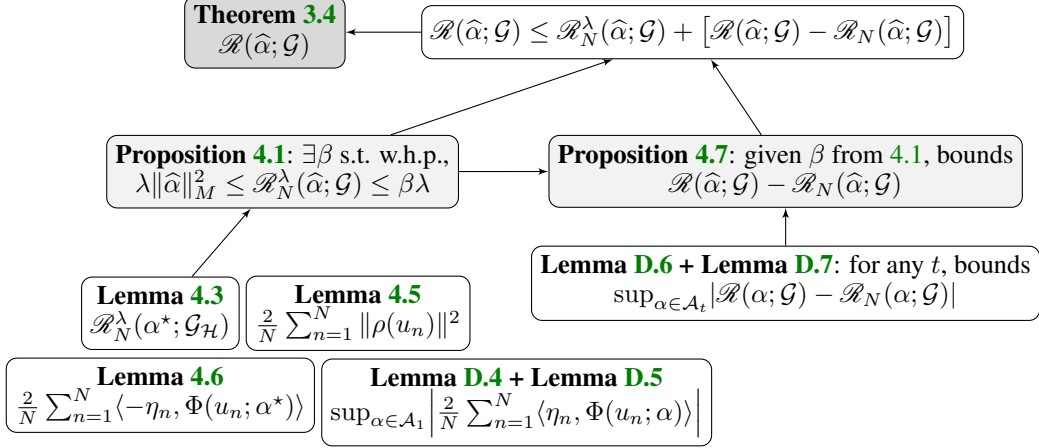

Figure 1: Flow chart illustrating the proof of Theorem 3.4.

## 4.1 Bounding the regularized empirical risk

The main technical contribution of this work is a tight bound on the $\mathcal{G}$-empirical risk $\mathscr{R}_N(\widehat{\alpha}; \mathcal{G})$ for the trained RFM. The analysis involves controlling several sources of error and careful truncation arguments to avoid unnecessarily strong assumptions on the problem. The result is the following.

**Proposition 4.1** (Regularized $\mathcal{G}$-empirical risk bound)**.** *Let Assumptions* 3.1 *and* 3.3 *hold. Suppose that* $\mathcal{G} = \rho + \mathcal{G}_\mathcal{H}$ *satisfies Assumption* 3.2. *Fix* $\delta \in (0, 1)$, $\lambda \in (0, 1)$, $M \in \mathbb{N}$, *and* $N \in \mathbb{N}$.

*Let $\widehat{\alpha} \in \mathbb{R}^M$ be the minimizer of the regularized $Y$-empirical risk $\mathscr{R}_N^\lambda(\,\cdot\,; Y)$ given by (2.3). If $M \geq \lambda^{-1} \log(16/\delta)$ and $N \geq \lambda^{-2} \log(8/\delta)$, then*

$$\mathscr{R}_N^\lambda(\widehat{\alpha}; \mathcal{G}) \leq \lambda \beta(\rho, \lambda, \mathcal{G}_\mathcal{H}, \eta) \tag{4.2}$$

*with probability at least $1 - \delta$, where the multiplicative factor $\beta(\rho, \lambda, \mathcal{G}_\mathcal{H}, \eta)$ is given by (3.3).*

Since $\lambda\|\widehat{\alpha}\|_M^2 \leq \mathscr{R}_N^\lambda(\widehat{\alpha}; \mathcal{G})$, the next corollary controlling the norm (2.3) of $\widehat{\alpha}$ is immediate. It plays a crucial role in developing an upper bound for the second term on the right side of (4.1).

**Corollary 4.2** (Trained RFM norm bound)**.** *Instantiate the hypotheses and notation of Prop. 4.1. Fix $\delta \in (0, 1)$ and $\lambda \in (0, 1)$. If $M \geq \lambda^{-1} \log(16/\delta)$ and $N \geq \lambda^{-2} \log(8/\delta)$, then*

$$\widehat{\alpha} \in \mathcal{A}_\beta := \left\{ \alpha \in \mathbb{R}^M \,\middle|\, \|\alpha\|_M^2 \leq \beta \right\} \tag{4.3}$$

*with probability at least $1 - \delta$. The radius $\beta := \beta(\rho, \lambda, \mathcal{G}_\mathcal{H}, \eta)$ of the norm bound is given by (3.3).*

The core elements of the proof of Prop. 4.1 are provided in the next few subsections, with the full argument in **SM** D.1. The main idea is to upper bound the $\mathcal{G}$-empirical risk by its regularized counterpart and then decompose the latter into several (coupled) error contributions.

To do this, first fix any $\alpha \in \mathbb{R}^M$. It holds that

$$\mathscr{R}_N^\lambda(\alpha; Y) = \mathscr{R}_N^\lambda(\alpha; \mathcal{G}) + \frac{2}{N}\sum_{n=1}^N \langle \eta_n, \mathcal{G}(u_n) - \Phi(u_n; \alpha) \rangle_\mathcal{Y} + \frac{1}{N}\sum_{n=1}^N \|\eta_n\|_\mathcal{Y}^2 \tag{4.4}$$

because $\mathcal{Y}$ is a Hilbert space and $y_n = \mathcal{G}(u_n) + \eta_n$. Using this, a short calculation shows that

$$\mathscr{R}_N^\lambda(\widehat{\alpha}; \mathcal{G}) = \left[ \mathscr{R}_N^\lambda(\widehat{\alpha}; Y) - \mathscr{R}_N^\lambda(\alpha; Y) \right] + \mathscr{R}_N^\lambda(\alpha; \mathcal{G}) + \frac{2}{N}\sum_{n=1}^N \langle \eta_n, \Phi(u_n; \widehat{\alpha}) - \Phi(u_n; \alpha) \rangle_\mathcal{Y}$$

$$\leq \mathscr{R}_N^\lambda(\alpha; \mathcal{G}) + \frac{2}{N}\sum_{n=1}^N \langle -\eta_n, \Phi(u_n; \alpha) \rangle_\mathcal{Y} + \frac{2}{N}\sum_{n=1}^N \langle \eta_n, \Phi(u_n; \widehat{\alpha}) \rangle_\mathcal{Y}. \tag{4.5}$$

In the second line, we used the fact that $\widehat{\alpha}$ minimizes $\mathscr{R}_N^\lambda(\,\cdot\,; Y)$. Since $\alpha \in \mathbb{R}^M$ is arbitrary, we have the freedom to choose $\alpha$ so that the first term is small (see Sect. 4.1.1 and 4.1.2). With $\alpha$ fixed, the second term averages to zero by our assumptions on the noise, and hence, we expect it to be small with high probability (see Sect. 4.1.3).

The third term in (4.5) exhibits high correlation between the noise $\eta_n$ and the trained RFM coefficients $\widehat{\alpha}$, making it more difficult to estimate. To control this last term, we first note that it is homogeneous in $\|\widehat{\alpha}\|_M$, which can be used to derive an upper bound in terms of a supremum over the unit ball with respect to $\|\cdot\|_M$. The resulting expression is then bounded with empirical process techniques (see **SM** D.1.3). For the complete details of the required argument we refer the reader to **SM** D.1.

In the remainder of this subsection, we estimate the first two terms on the right hand side of (4.5). Using the fact that $\mathcal{G} = \rho + \mathcal{G}_\mathcal{H}$, the first term can be split into two contributions,

$$\mathscr{R}_N^\lambda(\alpha; \mathcal{G}) \leq 2\mathscr{R}_N^\lambda(\alpha; \mathcal{G}_\mathcal{H}) + \frac{2}{N}\sum_{n=1}^N \|\rho(u_n)\|_\mathcal{Y}^2. \tag{4.6}$$

These contributions to the first term in (4.5) are bounded in Sect. 4.1.1 and 4.1.2. The second term in (4.5) is controlled in Sect. 4.1.3.

### 4.1.1 Bounding the approximation error

We begin with the term $\mathscr{R}_N^\lambda(\alpha; \mathcal{G}_\mathcal{H})$, which may be viewed as an empirical *approximation error* due to $\alpha$ being arbitrary. Its only dependence on the data is through $\{u_n\}$ in (2.3). Intuitively, this term should behave like its population counterpart. It is then natural to choose a Monte Carlo approximation $\alpha_m = \alpha_\mathcal{H}(\theta_m)$ for $\alpha$, where $\alpha_\mathcal{H} \in L_\mu^2(\Theta; \mathbb{R})$ is identified with $\mathcal{G}_\mathcal{H}$ as in (2.2). However, our intuition that $\lambda\|\alpha\|_M^2$ should concentrate around $\lambda\|\alpha_\mathcal{H}\|_{L_\mu^2}^2$ fails because it is generally not possible to control the tail of the random variable $|\alpha_\mathcal{H}(\theta)|^2$. We next show that this problem can be overcome by a carefully tuned truncation argument combined with Bernstein's inequality.

**Lemma 4.3** (Construction of approximator)**.** *Suppose that $\mathcal{G}_{\mathcal{H}} := \mathcal{G} \in \mathcal{H}$. Fix $\delta \in (0, 1)$, $\lambda > 0$, $N \in \mathbb{N}$, and $M \in \mathbb{N}$. Let $\{\theta_m\} \sim \mu^{\otimes M}$ and $\{u_n\} \sim \nu^{\otimes N}$. Define $\alpha^\star \in \mathbb{R}^M$ componentwise by*

$$\alpha_m^\star := \alpha_{\mathcal{H}}(\theta_m) \mathbb{1}_{\{|\alpha_{\mathcal{H}}(\theta_m)| \leq T\}}, \quad \text{where} \quad T := \sqrt{\lambda^{-1} \mathbb{E}_{\theta \sim \mu} |\alpha_{\mathcal{H}}(\theta)|^2} \tag{4.7}$$

*and $\mathcal{G}_{\mathcal{H}} = \mathbb{E}_{\theta \sim \mu}[\alpha_{\mathcal{H}}(\theta)\varphi(\,\cdot\,;\theta)]$ with $\|\mathcal{G}_{\mathcal{H}}\|_{\mathcal{H}}^2 = \mathbb{E}_{\theta \sim \mu}|\alpha_{\mathcal{H}}(\theta)|^2$. If $M \geq \lambda^{-1} \log(4/\delta)$, then with probability at least $1 - \delta$ in the random feature parameters $\theta_1, \ldots, \theta_M$, it holds that*

$$\mathscr{R}_N^\lambda(\alpha^\star; \mathcal{G}_{\mathcal{H}}) \leq 81\lambda \|\mathcal{G}_{\mathcal{H}}\|_{\mathcal{H}}^2. \tag{4.8}$$

**SM** D.1.1 provides the proof.

***Remark* 4.4** (Well-specified and noise-free)**.** *Lem.* 4.3 *gives a* $O(\lambda)$ *bound on the regularized* $\mathcal{G}_{\mathcal{H}}$*-empirical risk of a RFM trained on well-specified and noise-free iid data* $\{u_n, \mathcal{G}_{\mathcal{H}}(u_n)\}$*.*

#### 4.1.2 Bounding the misspecification error

The second contribution to (4.6) is easily bounded by Bernstein's inequality because $\rho \in L_\nu^\infty$. We refer the reader to **SM** D.1.2 for the detailed proof.

**Lemma 4.5** (Concentration of misspecification error)**.** *Let $\rho$ be as in Assumption* 3.2*. Fix $\delta \in (0, 1)$. With probability at least $1 - \delta$, it holds that*

$$\frac{2}{N} \sum_{n=1}^N \|\rho(u_n)\|_{\mathcal{Y}}^2 \leq 4 \mathbb{E}_{u \sim \nu} \|\rho(u)\|_{\mathcal{Y}}^2 + \frac{9\|\rho\|_{L_\nu^\infty}^2 \log(2/\delta)}{N}. \tag{4.9}$$

#### 4.1.3 Bounding the noise error

The second term on the right hand side of (4.5) is a zero mean error contribution due to the noise corrupting the output training data. By the fact that $\eta$ is subexponential (Assumption 3.3), Bernstein's inequality delivers exponential concentration. The proof details are in **SM** D.1.3.

**Lemma 4.6** (Concentration of noise error cross term)**.** *Let Assumptions* 3.1 *and* 3.3 *hold. Fix $\alpha \in \mathbb{R}^M$, $\{\theta_m\} \sim \mu^{\otimes M}$, and $\delta \in (0, 1)$. With probability at least $1 - \delta$, it holds that*

$$\frac{2}{N} \sum_{n=1}^N \langle -\eta_n, \Phi(u_n; \alpha) \rangle_{\mathcal{Y}} \leq 16e^{3/2} \|\eta_1\|_{\psi_1(\mathcal{Y})} \|\alpha\|_M \sqrt{\frac{\log(2/\delta)}{N}}. \tag{4.10}$$

**SM** D.1.3 also details the techniques used to upper bound the third and final term in (4.5).

### 4.2 Bounding the generalization gap

Having bounded the empirical risk with approximation arguments, it remains to control the estimation error $\mathscr{R}(\widehat{\alpha}; \mathcal{G}) - \mathscr{R}_N(\widehat{\alpha}; \mathcal{G})$ due to finite data in (4.1). We call this the *generalization gap*: the difference between the population test error and its empirical version. If $\widehat{\alpha}$ satisfies $\|\widehat{\alpha}\|_M^2 \leq t$ for some $t > 0$, then one can upper bound the generalization gap by its supremum over this set. The main challenge is to show existence of a (sufficiently small) $t$ that satisfies this inequality. This is handled by Cor. 4.2. As summarized in the following proposition, the resulting supremum of the empirical process defined by the generalization gap is shown to be of size $N^{-1/2}$ with high probability.

**Proposition 4.7** (Uniform bound on the generalization gap)**.** *Let Assumption* 3.1 *hold. Suppose $\mathcal{G}$ satisfies Assumption* 3.2*. Let $\{\theta_m\} \sim \mu^{\otimes M}$ for the RFM $\Phi$ given by* (2.1)*. Fix $\delta \in (0, 1)$. For iid input samples $\{u_n\} \sim \nu^{\otimes N}$, define the random variable*

$$\mathcal{E}_\beta(\{u_n\}, \{\theta_m\}) := \sup_{\alpha \in \mathcal{A}_\beta} \left| \frac{1}{N} \sum_{n=1}^N \|\mathcal{G}(u_n) - \Phi(u_n; \alpha)\|_{\mathcal{Y}}^2 - \mathbb{E}_u \|\mathcal{G}(u) - \Phi(u; \alpha)\|_{\mathcal{Y}}^2 \right|, \tag{4.11}$$

*where $\mathcal{A}_\beta := \{\alpha' \in \mathbb{R}^M \mid \|\alpha'\|_M^2 \leq \beta\}$ and the deterministic radius $\beta = \beta(\rho, \lambda, \mathcal{G}_{\mathcal{H}}, \eta)$ is given in* (3.3) *with $\mathcal{G}$ as above. If $N \geq \log(1/\delta)$, then with probability at least $1 - \delta$ it holds that*

$$\mathcal{E}_\beta(\{u_n\}, \{\theta_m\}) \leq 32e^{3/2} (\|\mathcal{G}\|_{L_\nu^\infty}^2 + \beta(\rho, \lambda, \mathcal{G}_{\mathcal{H}}, \eta)) \sqrt{\frac{6 \log(2/\delta)}{N}}. \tag{4.12}$$

The proof of Prop. 4.7 is given in **SM** D.2. The argument is composed of two steps. The first is to show that $\mathcal{E}_\beta \mid \{\theta_m\}$ concentrates around its (conditional) expectation (Lem. D.6). This follows easily using the boundedness of the summands. The second step is to upper bound the expectation of $\mathcal{E}_\beta \mid \{\theta_m\}$ (Lem. D.7). This is achieved by exploiting the Hilbert space structure of the $\mathcal{Y}$-square loss and the linearity of the RFM with respect to its coefficients.

### 4.3 Combining the bounds

Since we now have control over the $\mathcal{G}$-empirical risk and the generalization gap, the $\mathcal{G}$-population risk is also under control by (4.1). The proof of Thm. 3.4 follows by putting together the pieces (**SM** C).

## 5 Numerical experiment

To study how our theory holds up in practice, we numerically implement the vector-valued RF-RR algorithm on a benchmark operator learning dataset. The data $\{(u_n, \mathcal{G}(u_n))\}_{n=1}^N$ is noise-free, the $\{u_n\}$ are iid Gaussian random fields, and $\mathcal{G}\colon L^2(\mathbb{T}; \mathbb{R}) \to L^2(\mathbb{T}; \mathbb{R})$ is a nonlinear operator defined as the time one flow map of the viscous Burgers' equation on the torus $\mathbb{T}$. **SM** E provides more details about the problem setting and the choice of random feature pair $(\varphi, \mu)$.

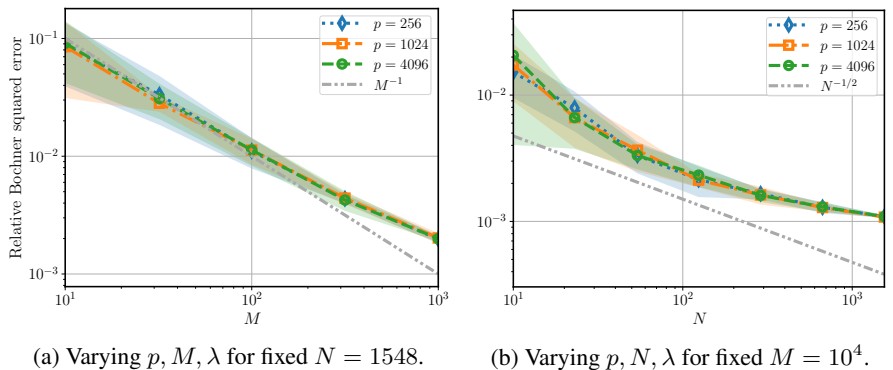

(a) Varying $p, M, \lambda$ for fixed $N = 1548$.  (b) Varying $p, N, \lambda$ for fixed $M = 10^4$.

Figure 2: Squared test error of trained RFM for learning the Burgers' equation solution operator. All shaded bands denote two empirical standard deviations from the empirical mean of the error computed over 10 different models, each with iid sampling of the features and training data indices.

Figure 2a shows the decay of the relative squared test error as $M$ increases (with $\lambda \simeq 1/M$) for fixed $N$. The error closely follows the rate $O(M^{-1})$ until it begins to saturate at larger $M$. This is due to either $\mathcal{G}$ not belonging to the RKHS of $(\varphi, \mu)$ or the finite data error dominating. As implied by our theory, the error does not depend on the discretized output dimension $p < \infty$. Figure 2b displays similar behavior as $N$ is varied (now with $\lambda \simeq 1/\sqrt{N}$ and fixed $M$). Overall, the observed parameter and sample complexity reasonably validate our theoretical insights.

## 6 Conclusion

This paper establishes several fundamental results for learning with infinite-dimensional vector-valued random features; these include strong consistency and explicit convergence rates. When the underlying mapping belongs to the RKHS, the rates obtained in this work match minimax optimal rates in the number of samples $N$, requiring only a number of random features $M \simeq \sqrt{N}$. Despite being derived in a very general setting, to the best of our knowledge, this provides the sharpest parameter complexity in $M$, which is free from logarithmic factors for the first time.

There are several interesting directions for future work. These include deriving fast rates, relaxing the boundedness assumption on the features and the true mapping, and accommodating heavier-tailed or white noise distributions. Obtaining fast rates would require a sharpening of several estimates, and in particular, replacing the global Rademacher complexity-type estimate, implicit in our work, by its local counterpart. As our approach does not make use of an explicit solution formula, which is only available for a square loss, this might pave the way for improved rates for other loss functions, such as a general $L^p$-loss. We leave such potential extensions of the present approach for future work.

## Acknowledgments and Disclosure of Funding

The work of SL is supported by Postdoc.Mobility grant P500PT-206737 from the Swiss National Science Foundation. NHN acknowledges support from the National Science Foundation Graduate Research Fellowship Program under award number DGE-1745301 and from the Amazon/Caltech AI4Science Fellowship, and partial support from the Air Force Office of Scientific Research under MURI award number FA9550-20-1-0358 (Machine Learning and Physics-Based Modeling and Simulation). SL and NHN are also grateful for partial support from the Department of Defense Vannevar Bush Faculty Fellowship held by Andrew M. Stuart under Office of Naval Research award number N00014-22-1-2790.

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

# A  Concentration of measure

In this section, we recall two classical results from [35] that estimate the difference between empirical averages and true averages of random vectors taking values in Banach spaces. These are then specialized to the setting of subexponential random variables, which play a major role in this paper. To set the notation, we use $\Pr$ to denote probability with respect to the underlying probability space.

The first result is a vector-valued Bernstein concentration inequality with various applications to problems posed in infinite-dimensional Hilbert spaces [6, 28, 38]. It is used throughout this work.

**Theorem A.1** (Vector-valued Bernstein inequality in Hilbert space). *Let $Z$ be an $H$-valued random variable, where $(H, \langle \cdot, \cdot \rangle, \| \cdot \|)$ is a separable Hilbert space. Suppose there exist positive numbers $b > 0$ and $\sigma > 0$ such that*

$$\mathbb{E}\|Z - \mathbb{E}\,Z\|^p \leq \frac{1}{2}p!\sigma^2 b^{p-2} \quad \text{for all} \quad p \geq 2\,. \tag{A.1}$$

*For any $\delta \in (0, 1)$ and $N \in \mathbb{N}$, denoting by $\{Z_n\}_{n=1}^N$ a sequence of $N$ iid copies of $Z$, it holds that*

$$\Pr\left\{ \left\| \frac{1}{N}\sum_{n=1}^N Z_n - \mathbb{E}\,Z \right\| \leq \frac{2b\log(2/\delta)}{N} + \sqrt{\frac{2\sigma^2\log(2/\delta)}{N}} \right\} \geq 1 - \delta\,. \tag{A.2}$$

*Proof.* The result is a direct consequence of [35, Cor. 1, p. 144] in the iid Hilbert space setting. In this case, it holds for any $t > 0$ that

$$\Pr\{\|S_N - \mathbb{E}\,S_N\| \geq t\} \leq 2\exp\left(-\frac{N^2 t^2}{2N\sigma^2 + 2Ntb}\right) = 2\exp\left(-\frac{Nt^2}{2\sigma^2 + 2bt}\right),$$

where $S_n := \frac{1}{N}\sum_{n=1}^N Z_n$. Setting the right hand side equal to $\delta$, solving a quadratic equation for $t = t(\delta)$, and using the inequality $\sqrt{a+b} \leq \sqrt{a} + \sqrt{b}$ to bound $t(\delta)$ from above leads to (A.2). $\square$

The most common use case of Bernstein's inequality is in the following bounded setting.

**Lemma A.2.** *Let $Z$ be a (potentially) uncentered random variable such that*

$$\|Z\| \leq c \quad \text{almost surely} \quad \text{and} \quad \mathbb{E}\|Z - \mathbb{E}\,Z\|^2 \leq v^2 \tag{A.3}$$

*for some $c > 0$ and $v > 0$. Then $Z$ satisfies Bernstein's moment condition (A.1) with $b = 2c$ and $\sigma = v$. If $\mathbb{E}\,Z = 0$, then taking $b = c$ suffices.*

*Proof.* It holds that $\|Z - \mathbb{E}\,Z\| \leq \|Z\| + \mathbb{E}\|Z\| \leq 2c$ almost surely. We compute

$$\mathbb{E}\|Z - \mathbb{E}\,Z\|^p = \mathbb{E}[\|Z - \mathbb{E}\,Z\|^2\|Z - \mathbb{E}\,Z\|^{p-2}] \leq \mathbb{E}\|Z - \mathbb{E}\,Z\|^2 (2c)^{p-2}$$

$$\leq v^2(2c)^{p-2} \leq \frac{1}{2}p!v^2(2c)^{p-2}$$

because $1 \leq p!/2$ for all $p \geq 2$. The centered improvement is trivial. $\square$

The second classic result we present is a one-sided Bernstein-type tail bound in a general Banach space. We invoke this theorem to control the tails of suprema of empirical processes.

**Theorem A.3** (Vector-valued Bernstein inequality in Banach space). *Let $Z$ be a $\mathcal{Z}$-valued random variable, where $(\mathcal{Z}, \| \cdot \|)$ is a separable Banach space. Suppose there exist positive numbers $b > 0$ and $\sigma > 0$ such that*

$$\mathbb{E}\|Z - \mathbb{E}\,Z\|^p \leq \frac{1}{2}p!\sigma^2 b^{p-2} \quad \text{for all} \quad p \geq 2\,. \tag{A.4}$$

*For any $\delta \in (0, 1)$ and $N \in \mathbb{N}$, denoting by $\{Z_n\}_{n=1}^N$ a sequence of $N$ iid copies of $Z$, it holds that*

$$\Pr\left\{ \left\| \frac{1}{N}\sum_{n=1}^N Z_n \right\| - \mathbb{E}\left\| \frac{1}{N}\sum_{n=1}^N Z_n \right\| \leq \frac{2b\log(1/\delta)}{N} + \sqrt{\frac{2\sigma^2\log(1/\delta)}{N}} \right\} \geq 1 - \delta\,. \tag{A.5}$$

*Proof.* The assertion is [35, Cor. 1, p. 144] in the iid Banach space setting (hence only convergence of norms in (A.5) instead of strong convergence). It is proved the same way as Thm. A.1. □

In Theorem A.3, a random variable $Z$ satisfying the *Bernstein moment condition* (A.4) is subexponential in the sense that $\|Z - \mathbb{E}\,Z\|$ is subexponential on $\mathbb{R}$, i.e., exhibits exponential tail decay. Recall that for a real-valued random variable $Z$, its subexponential norm may be defined by

$$\|Z\|_{\psi_1} := \sup_{p \in [1,\infty)} \frac{(\mathbb{E}|Z|^p)^{1/p}}{p}\,. \tag{A.6}$$

See [45, Sect. 2.7] for equivalent definitions. We say that $Z$ is subexponential if its subexponential norm is finite. Following [31, Sect. 4.3, pp. 19–20], for a random variable $Z$ with values in a Banach space $(\mathcal{Z}, \|\cdot\|_{\mathcal{Z}})$ we define

$$\|Z\|_{\psi_1(\mathcal{Z})} := \big\|\|Z\|_{\mathcal{Z}}\big\|_{\psi_1} = \sup_{p \in [1,\infty)} \frac{(\mathbb{E}\|Z\|_{\mathcal{Z}}^p)^{1/p}}{p} \tag{A.7}$$

as its subexponential norm. It is known that a random variable has finite subexponential norm if and only if it satisfies the Bernstein moment condition (A.4) [see, e.g., 31, Appendix A.2]. Next, we give explicit constants in the Bernstein moment condition that depend on the subexponential norm.

**Proposition A.4** (Subexponential implies Bernstein moment condition)**.** *Let $Z$ be a $(\mathcal{Z}, \|\cdot\|)$-valued subexponential random variable, that is, $\|Z\|_{\psi_1(\mathcal{Z})} < \infty$. Then $Z$ satisfies*

$$\mathbb{E}\|Z - \mathbb{E}\,Z\|^p \le \frac{1}{2} p! \sigma^2 b^{p-2} \quad \text{for all} \quad p \ge 2\,, \tag{A.8}$$

*where*

$$\sigma^2 := 4e\sqrt{\mathbb{E}\|Z - \mathbb{E}\,Z\|^2}\|Z\|_{\psi_1(\mathcal{Z})} \quad \text{and} \quad b := 4e\|Z\|_{\psi_1(\mathcal{Z})}\,. \tag{A.9}$$

*Proof.* By the Cauchy–Schwarz inequality,

$$\mathbb{E}\|Z - \mathbb{E}\,Z\|^p = \mathbb{E}[\|Z - \mathbb{E}\,Z\|\|Z - \mathbb{E}\,Z\|^{p-1}] \le \|Z - \mathbb{E}\,Z\|_{L^2_{\mathbb{P}}}(\mathbb{E}\|Z - \mathbb{E}\,Z\|^{2p-2})^{1/2}\,.$$

The inequality $|a + b|^q \le 2^{q-1}(|a|^q + |b|^q)$, Jensen's inequality, and $\|Z\|_{\psi_1(\mathcal{Z})} < \infty$ show that

$$\mathbb{E}\|Z - \mathbb{E}\,Z\|^{2p-2} \le 2^{2p-2}\,\mathbb{E}\|Z\|^{2p-2} \le 2^{2p-2}(2p-2)^{2p-2}\|Z\|_{\psi_1(\mathcal{Z})}^{2p-2}\,.$$

Next, Stirling's approximation $(q/e)^q \le q!$ and $q! = q(q-1)! \ge 2(q-1)!$ for $q \ge 2$ yields

$$\mathbb{E}\|Z - \mathbb{E}\,Z\|^p \le 2^{2p-2}\|Z - \mathbb{E}\,Z\|_{L^2_{\mathbb{P}}}(p-1)^{p-1}\|Z\|_{\psi_1(\mathcal{Z})}^{p-1}$$

$$\le (p!/2)\|Z - \mathbb{E}\,Z\|_{L^2_{\mathbb{P}}}2^{2p-2}e^{p-1}\|Z\|_{\psi_1(\mathcal{Z})}^{p-1}\,.$$

Rearranging the exponents to fit the Bernstein moment condition form completes the proof. □

This leads to the following corollary, which is useful in the setting that the variance $\mathbb{E}\|Z - \mathbb{E}\,Z\|_{\mathcal{Z}}^2 = \|Z - \mathbb{E}\,Z\|_{L^2_{\mathbb{P}}(\Omega;\mathcal{Z})}^2$ of random variable $Z$ is not small or hard to compute.

**Corollary A.5** (Subexponential tail bound in Banach space)**.** *Fix $N \in \mathbb{N}$. Let $\{Z_n\}_{n=1}^N$ be iid random variables with values in a separable Banach space $(\mathcal{Z}, \|\cdot\|)$. Suppose that $\|Z_1\|_{\psi_1(\mathcal{Z})} < \infty$. Let $S_N := \frac{1}{N}\sum_{n=1}^N Z_n$. Fix $\delta \in (0, 1)$. With probability at least $1 - \delta$, it holds that*

$$\big\|S_N\big\| - \mathbb{E}\big\|S_N\big\| \le \frac{8e\|Z_1\|_{\psi_1(\mathcal{Z})}\log(1/\delta)}{N} + \sqrt{\frac{8e\|Z_1 - \mathbb{E}\,Z_1\|_{L^2_{\mathbb{P}}(\Omega;\mathcal{Z})}\|Z_1\|_{\psi_1(\mathcal{Z})}\log(1/\delta)}{N}}\,. \tag{A.10}$$

*In particular, if $N \ge \log(1/\delta)$, then with probability at least $1 - \delta$ it holds that*

$$\big\|S_N\big\| - \mathbb{E}\big\|S_N\big\| \le \sqrt{\frac{64e^3\|Z_1\|_{\psi_1(\mathcal{Z})}^2\log(1/\delta)}{N}}\,. \tag{A.11}$$

*Proof.* Apply Prop. A.4 to Thm. A.3 to obtain the first assertion. For the second assertion, first note that $\mathbb{E}\|Z_1 - \mathbb{E}\,Z_1\|^2 \leq 4\,\mathbb{E}\|Z_1\|^2$ (by triangle inequality and using $(a+b)^2 \leq 2(a^2+b^2)$) and $\|Z_1\|^2_{\psi_1(\mathcal{Z})} \geq \mathbb{E}\|Z_1\|^2/4$ (by (A.7)). Since $N \geq \log(1/\delta)$, we have $\log(1/\delta)/N \leq \sqrt{\log(1/\delta)/N}$. Combining these facts, it follows that the right hand side of (A.10) is bounded above by

$$\sqrt{\frac{64e^2\|Z_1\|^2_{\psi_1(\mathcal{Z})}\log(1/\delta)}{N}} + \sqrt{\frac{32e\|Z_1\|^2_{\psi_1(\mathcal{Z})}\log(1/\delta)}{N}} \leq \sqrt{\frac{64(2e^2+e)\|Z_1\|^2_{\psi_1(\mathcal{Z})}\log(1/\delta)}{N}}.$$

We used $\sqrt{a} + \sqrt{b} \leq \sqrt{2(a+b)}$ on the right. Noting that $2e^2 + e \leq e^3$ completes the proof. $\qquad\square$

Comparing this result to a similar result [28, Prop. 7(i), pp. 4–5, the iid case], we note that (A.10) in Cor. A.5 is sharper in the sense that the constant in the $N^{-1/2}$ term depends on the variance of the summands instead of just its subexponential norm (which can be much larger than the variance).

# B  Misspecification error with RKHS methods

Let $K\colon \mathcal{X} \times \mathcal{X} \to \mathcal{L}(\mathcal{Y})$ be an operator-valued kernel [32] corresponding to a separable[2] RKHS $\mathcal{H}$ of functions mapping $\mathcal{X}$ to $\mathcal{Y}$. Here, $\mathcal{L}(\mathcal{Y})$ denotes the set of bounded linear operators from $\mathcal{Y}$ into itself. In this section, we present a general analysis based on [42] for the approximation of elements in $L^2_\nu(\mathcal{X}; \mathcal{Y})$ by elements in the RKHS $\mathcal{H}$. This problem is relevant to our learning theory framework whenever the true data-generating map does not belong to the RKHS (2.2) associated to the given random feature pair $(\varphi, \mu)$. Specializing to this setting, suppose that $(\varphi, \mu)$ satisfy Assumption 3.1. Let $K\colon (u, u') \mapsto \mathbb{E}_{\theta \sim \mu}[\varphi(u; \theta) \otimes_\mathcal{Y} \varphi(u'; \theta)]$ be the corresponding limit random feature kernel. It holds that $K(u, u)$ is trace-class for each $u \in \mathcal{X}$ $\nu$-almost surely because

$$\mathrm{tr}(K(u, u)) = \mathbb{E}_\theta\,\mathrm{tr}(\varphi(u; \theta) \otimes_\mathcal{Y} \varphi(u; \theta)) = \mathbb{E}_\theta\|\varphi(u; \theta)\|^2_\mathcal{Y} \leq \|\varphi\|^2_{L^\infty} < \infty \tag{B.1}$$

by the Fubini–Tonelli theorem. It follows from [8, Prop. 4.8, p. 394] that $\mathcal{H}$ is compactly embedded into $L^2_\nu(\mathcal{X}; \mathcal{Y})$ because

$$\mathbb{E}_{u \sim \nu}\|K(u, u)\|_{\mathcal{L}(\mathcal{Y})} \leq \mathbb{E}_{u \sim \nu}\,\mathrm{tr}(K(u, u)) \leq \|\varphi\|^2_{L^\infty} < \infty\,. \tag{B.2}$$

We denote the canonical embedding by $\iota\colon \mathcal{H} \hookrightarrow L^2_\nu(\mathcal{X}; \mathcal{Y})$. Now let $\mathcal{G} \in L^2_\nu(\mathcal{X}; \mathcal{Y})$ be an arbitrary operator. We consider an approximation $\mathcal{G}_\vartheta$ to $\mathcal{G}$ defined by

$$\mathcal{G}_\vartheta := \operatorname*{arg\,min}_{F \in \mathcal{H}}\Big\{\|\mathcal{G} - \iota F\|^2_{L^2_\nu} + \vartheta\|F\|^2_\mathcal{H}\Big\}\,. \tag{B.3}$$

This operator has a simple representation.

**Lemma B.1.** *There exists a unique solution $\mathcal{G}_\vartheta$ to* (B.3) *given by*

$$\mathcal{G}_\vartheta = (\iota^*\iota + \vartheta\,\mathrm{Id})^{-1}\iota^*\mathcal{G}\,. \tag{B.4}$$

*Proof.* The result is a consequence of convex optimization on Hilbert spaces and the fact that $\iota$ is a bounded linear operator. $\qquad\square$

The adjoint of the inclusion map, $\iota^*\colon L^2_\nu(\mathcal{X}; \mathcal{Y}) \to \mathcal{H}$, is given by the vector-valued reproducing kernel property as the (Bochner) integral operator

$$F \mapsto \iota^*F = \mathbb{E}_{u \sim \nu}[K(\,\cdot\,, u)F(u)]\,. \tag{B.5}$$

Since $\iota$ is compact, so is $\mathcal{K} := \iota\iota^* \in \mathcal{L}(L^2_\nu(\mathcal{X}; \mathcal{Y}))$. The action of $\mathcal{K}$ is the same as that of the integral operator $\iota^*$ above. Since $\mathcal{K}$ is also symmetric, the spectral theorem yields the operator norm convergent expansion $\mathcal{K} = \sum_j \lambda_j e_j \otimes_{L^2(\nu)} e_j$. The sequence $\{\lambda_j\} \subset \mathbb{R}_{\geq 0}$ is a non-increasing rearrangement of the eigenvalues of $\mathcal{K}$ and $\{e_j\}$ is its corresponding eigenbasis.

Now define the regularized RKHS approximation error

$$\mathscr{A}_\mathcal{G}(\vartheta) := \inf_{F \in \mathcal{H}}\Big\{\|\mathcal{G} - \iota F\|^2_{L^2_\nu} + \vartheta\|F\|^2_\mathcal{H}\Big\} \tag{B.6}$$

which is parametrized by $\mathcal{G} \in L^2_\nu(\mathcal{X}; \mathcal{Y})$. We have the following convergence result for this error.

---

[2]The *assumption* of separability of the RKHS could be removed if additional conditions are placed on its kernel that would *imply* separability, the most relevant being continuity-type assumptions. In the case $\mathcal{Y} = \mathbb{R}$, it is known that existence of a Borel measurable feature map for the kernel suffices for separability of its RKHS [34], which is much weaker than continuity. However, we are unaware of similar results for general $\mathcal{Y}$.

**Lemma B.2** (Convergence of regularized RKHS approximation error). *Suppose that $\mathcal{G}$ is in the $L_\nu^2$-closure of $\mathcal{H}$. Under the prevailing assumptions of this section,*

$$\lim_{\vartheta \to 0} \mathscr{A}_{\mathcal{G}}(\vartheta) = 0\,. \tag{B.7}$$

*Proof.* By Lem. B.1 and the Woodbury identity [33, Thm. 1], it holds that

$$\iota\mathcal{G}_\vartheta = \iota(\iota^*\iota + \vartheta\,\mathrm{Id}_{\mathcal{H}})^{-1}\iota^*\mathcal{G} = \mathcal{K}(\mathcal{K} + \vartheta\,\mathrm{Id}_{L_\nu^2})^{-1}\mathcal{G} = (\mathcal{K} + \vartheta\,\mathrm{Id}_{L_\nu^2})^{-1}\mathcal{K}\mathcal{G}\,.$$

The second equality holds by simultaneous diagonalization. Writing $\mathcal{G}$ in the eigenbasis of $\mathcal{K}$ yields

$$\mathcal{G} - \iota\mathcal{G}_\vartheta = \big[\,\mathrm{Id}_{L_\nu^2} - (\mathcal{K} + \vartheta\,\mathrm{Id}_{L_\nu^2})^{-1}\mathcal{K}\big]\mathcal{G} = \sum_{j \in \mathbb{N}} \frac{\vartheta}{\lambda_j + \vartheta}\langle e_j, \mathcal{G}\rangle_{L_\nu^2}\, e_j\,.$$

Similarly, using the norm isometry between $L_\nu^2$ and the RKHS [see, e.g., 8, pp. 403–404] we obtain

$$\|\mathcal{G}_\vartheta\|_{\mathcal{H}}^2 = \|\mathcal{K}^{-1/2}\iota\mathcal{G}_\vartheta\|_{L_\nu^2}^2 = \|\mathcal{K}^{1/2}(\mathcal{K} + \vartheta\,\mathrm{Id}_{L_\nu^2})^{-1}\mathcal{G}\|_{L_\nu^2}^2 = \sum_{j \in \mathbb{N}} \left(\frac{\sqrt{\lambda_j}}{\lambda_j + \vartheta}\right)^2 \langle e_j, \mathcal{G}\rangle_{L_\nu^2}^2\,.$$

Since the infimum in (B.6) is attained at $\mathcal{G}_\vartheta$ (B.3), we deduce that

$$\begin{aligned}
\mathscr{A}_{\mathcal{G}}(\vartheta) &= \|\mathcal{G} - \iota\mathcal{G}_\vartheta\|_{L_\nu^2}^2 + \vartheta\|\mathcal{G}_\vartheta\|_{\mathcal{H}}^2 \\
&= \sum_{j \in \mathbb{N}} \frac{\vartheta^2}{(\lambda_j + \vartheta)^2}\langle e_j, \mathcal{G}\rangle_{L_\nu^2}^2 + \sum_{j \in \mathbb{N}} \frac{\vartheta\lambda_j}{(\lambda_j + \vartheta)^2}\langle e_j, \mathcal{G}\rangle_{L_\nu^2}^2 \\
&= \sum_{j \in \mathbb{N}} \left(\frac{\vartheta}{\lambda_j + \vartheta}\right)\langle e_j, \mathcal{G}\rangle_{L_\nu^2}^2\,.
\end{aligned}$$

Using $\vartheta/(\lambda_j + \vartheta) \le 1$ for each $j \in \mathbb{N}$ and $\mathcal{G} \in \overline{\mathcal{H}}^{L_\nu^2}$ (the $L_\nu^2$-closure of $\mathcal{H}$), it follows that

$$\mathscr{A}_{\mathcal{G}}(\vartheta) = \sum_{\{j \in \mathbb{N}\,|\,\lambda_j \ne 0\}} \left(\frac{\vartheta}{\lambda_j + \vartheta}\right)\langle e_j, \mathcal{G}\rangle_{L_\nu^2}^2 \to 0 \quad \text{as} \quad \vartheta \to 0 \tag{B.8}$$

by dominated convergence. $\qquad\square$

The *rate* of convergence of $\mathscr{A}_{\mathcal{G}}$ to zero can be quantified under an additional regularity assumption.

**Lemma B.3** (Convergence rate under Hölder source condition). *Suppose $\mathcal{G} \in \mathrm{Im}(\mathcal{K}^{r/2})$ for some $r \ge 0$. Then for any $\vartheta > 0$, it holds under the prevailing assumptions of this section that*

$$\mathscr{A}_{\mathcal{G}}(\vartheta) \le \|\mathcal{K}^{-r/2}\mathcal{G}\|_{L_\nu^2(\mathcal{X};\mathcal{Y})}^2 \times \begin{cases} \vartheta^r, & \text{if } r \in [0,1]\,, \\ \vartheta\|\mathcal{K}\|_{\mathcal{L}(L_\nu^2)}^{r-1}, & \text{if } r > 1\,. \end{cases} \tag{B.9}$$

*Proof.* The proof closely follows the argument of Smale and Zhou [42, Thm. 4, p. 295]. By hypothesis, there exists $F_{\mathcal{G}} \in L_\nu^2(\mathcal{X};\mathcal{Y})$ such that $\mathcal{G} = \mathcal{K}^{r/2}F_{\mathcal{G}}$. Then

$$\mathscr{A}_{\mathcal{G}}(\vartheta) = \sum_{j \in \mathbb{N}} \frac{\vartheta\lambda_j^r}{\lambda_j + \vartheta}\langle e_j, F_{\mathcal{G}}\rangle_{L_\nu^2}^2 \le \left(\sup_{j \in \mathbb{N}} \frac{\vartheta\lambda_j^r}{\lambda_j + \vartheta}\right)\|F_{\mathcal{G}}\|_{L_\nu^2}^2\,.$$

For any $j \in \mathbb{N}$, the argument of the supremum equals

$$\frac{\vartheta\lambda_j^r}{\lambda_j + \vartheta} = \left(\frac{\lambda_j}{\lambda_j + \vartheta}\right)\left(\frac{\lambda_j}{\lambda_j + \vartheta}\right)^{r-1}\frac{\vartheta}{(\lambda_j + \vartheta)^{1-r}} = \vartheta^r\left(\frac{\lambda_j}{\lambda_j + \vartheta}\right)^r\left(\frac{\vartheta}{\lambda_j + \vartheta}\right)^{1-r}\,.$$

This is bounded above by $\vartheta^r$ for all $j \in \mathbb{N}$ if $r \ge 0$ and $r \le 1$. Otherwise, $r > 1$ and

$$\frac{\vartheta\lambda_j^r}{\lambda_j + \vartheta} = \vartheta\lambda_j^{r-1}\left(\frac{\lambda_j}{\lambda_j + \vartheta}\right) \le \vartheta\lambda_j^{r-1}\,.$$

Taking the supremum over $j \in \mathbb{N}$ completes the proof. $\qquad\square$

In Lem. B.3, $\mathcal{G}$ satisfies $\mathcal{G} \in \mathcal{H}$ if $r \ge 1$, and the rate of convergence of $\mathscr{A}_{\mathcal{G}}(\vartheta)$ is at least as fast as $O(\vartheta)$ as $\vartheta \to 0$. When $r \in [0,1)$, then $\mathcal{G} \notin \mathcal{H}$ and the rate becomes slower than linear in $\vartheta$.

# C    Proofs for section 3

In this section, we prove Thm. 3.4 and its main consequences.

*Proof of Theorem 3.4.* Under the hypotheses, (4.2) in Prop. 4.1 holds with probability at least $1 - \delta$ provided that $M \geq \lambda^{-1} \log(16/\delta)$ and $N \geq \lambda^{-2} \log(8/\delta)$. That is, $\mathscr{R}_N(\widehat{\alpha}; \mathcal{G}) \leq \mathscr{R}_N^\lambda(\widehat{\alpha}; \mathcal{G}) \leq \beta\lambda$, where $\beta = \beta(\rho, \lambda, \mathcal{G}_\mathcal{H}, \eta)$ is given by (3.3). In particular, $\widehat{\alpha} \in \mathcal{A}_\beta$ (4.3) on the same event (by Cor. 4.2). It follows that, on this event,

$$\mathscr{R}(\widehat{\alpha}; \mathcal{G}) - \mathscr{R}_N(\widehat{\alpha}; \mathcal{G}) \leq \sup_{\alpha \in \mathcal{A}_\beta} \left| \mathscr{R}_N(\alpha; \mathcal{G}) - \mathscr{R}(\alpha; \mathcal{G}) \right|.$$

Application of Prop. 4.7 shows that the right hand side of the above display is bounded above by

$$32\sqrt{6}e^{3/2}\big(\|\mathcal{G}\|_{L_\nu^\infty}^2 + \|\varphi\|_{L^\infty}^2 \beta(\rho, \lambda, \mathcal{G}_\mathcal{H}, \eta)\big)\lambda \leq 79e^{3/2}\big(\|\mathcal{G}\|_{L_\nu^\infty}^2 + \|\varphi\|_{L^\infty}^2 \beta(\rho, \lambda, \mathcal{G}_\mathcal{H}, \eta)\big)\lambda$$

with probability at least $1 - \delta$ because $N \geq \lambda^{-2}\log(8/\delta) \geq \log(2/\delta)$. Recalling (4.1), using $1 \leq 79e^{3/2}$, and applying a union bound completes the proof. $\qquad\square$

*Proof of Corollary 3.8.* Since $\mathcal{G} = \rho + \mathcal{G}_\mathcal{H}$, we compute

$$\begin{aligned}
\mathscr{R}(\widehat{\alpha}; \mathcal{G}_\mathcal{H}) = \big\|\mathcal{G}_\mathcal{H} - \Phi(\,\cdot\,; \widehat{\alpha})\big\|_{L_\nu^2}^2 &= \big\|-\rho + [\mathcal{G} - \Phi(\,\cdot\,; \widehat{\alpha})]\big\|_{L_\nu^2}^2 \\
&\leq 2\|\rho\|_{L_\nu^2}^2 + 2\|\mathcal{G} - \Phi(\,\cdot\,; \widehat{\alpha})\|_{L_\nu^2}^2 \\
&= 2\,\mathbb{E}_{u \sim \nu}\|\rho(u)\|_\mathcal{Y}^2 + 2\mathscr{R}(\widehat{\alpha}; \mathcal{G}).
\end{aligned}$$

By (3.2) and (3.3), we see that the term $\mathbb{E}\|\rho(u)\|_\mathcal{Y}^2$ also appears in the upper bound for $\mathscr{R}(\widehat{\alpha}; \mathcal{G})$. Collecting like terms and enlarging constants proves the assertion. $\qquad\square$

*Proof of Theorem 3.10.* For $k \in \mathbb{N}$ and $\delta \in (0, 1)$, the trained RFM satisfies $\|\mathcal{G} - \Phi(\,\cdot\,; \widehat{\alpha}^{(k)})\|_{L_\nu^2} \leq cs_k$ for some deterministic constant $c > 0$, where the sequence $s_n \to 0$ as $n \to \infty$ is given by the right hand side of (3.9) with $\lambda = \lambda_n$ for each $n \in \mathbb{N}$ (by Lem. B.2). This inequality holds with probability at least $1 - \delta$ if $M \gtrsim \lambda_k^{-1} \log(2/\delta)$ and $N \gtrsim \lambda_k^{-2} \log(2/\delta)$ by Thm. 3.4. Now choose $\delta = \lambda_k$. Then

$$\sum_{k \in \mathbb{N}} \Pr\big\{\|\mathcal{G} - \Phi(\,\cdot\,; \widehat{\alpha}^{(k)})\|_{L_\nu^2} > cs_k\big\} \leq \sum_{k \in \mathbb{N}} \lambda_k < \infty.$$

The first Borel–Cantelli lemma establishes that there exists an $\mathbb{N}$-valued random variable $k_0$ such that $\|\mathcal{G} - \Phi(\,\cdot\,; \widehat{\alpha}^{(k)})\|_{L_\nu^2} \leq cs_k$ for all $k > k_0$ almost surely. This implies the desired result. $\qquad\square$

*Proof of Theorem 3.12.* Application of Thm. 3.4 leads to the high probability bound (3.9) by the same argument from Sect. 3.3. Using that $\lambda \lesssim 1$ and Lem. B.3 proves the assertion. $\qquad\square$

*Proof of Corollary 3.13.* Apply Thm. 3.12 to get a high probability bound. Choose $\lambda = \lambda_k = \delta$. Then the proof follows that of Thm. 3.10 except with $\{s_k\}$ replaced by $\{\lambda_k^{(r \wedge 1)/2}\}$. $\qquad\square$

# D    Proofs for section 4

This section provides proofs of the error bounds for the regularized empirical risk (**SM** D.1) and the generalization gap (**SM** D.2).

## D.1    Proofs for subsection 4.1: Bounding the regularized empirical risk

We now prove Prop. 4.1. Supporting results used in the proof appear after the argument in the subsequent subsections (**SM** D.1.1, D.1.2, and D.1.3).

*Proof of Proposition 4.1.* Our starting point is (4.5). We first note by linearity that

$$\frac{2}{N}\sum_{n=1}^{N}\langle\eta_n,\Phi(u_n;\widehat{\alpha})\rangle_{\mathcal{Y}} = \|\widehat{\alpha}\|_M\left(\frac{2}{N}\sum_{n=1}^{N}\langle\eta_n,\Phi(u_n;\widehat{\alpha}/\|\widehat{\alpha}\|_M)\rangle_{\mathcal{Y}}\right)$$

$$\leq \|\widehat{\alpha}\|_M\left(2\sup_{\alpha'\in\mathcal{A}_1}\left|\frac{1}{N}\sum_{n=1}^{N}\langle\eta_n,\Phi(u_n;\alpha')\rangle_{\mathcal{Y}}\right|\right),$$

where $\mathcal{A}_1 = \left\{\alpha'\in\mathbb{R}^M \,\middle|\, \|\alpha'\|_M^2 \leq 1\right\}$, provided that $\|\widehat{\alpha}\|_M > 0$. Otherwise, the inequality in the above display holds trivially. Next, we define

$$t := \frac{1}{M}\sum_{m=1}^{M}|\widehat{\alpha}_m|^2 = \|\widehat{\alpha}\|_M^2\,, \tag{D.1}$$

$$A_{N,M}^\lambda := \mathscr{R}_N^\lambda(\alpha;\mathcal{G}) + \frac{2}{N}\sum_{n=1}^{N}\langle-\eta_n,\Phi(u_n;\alpha)\rangle_{\mathcal{Y}}\,, \quad\text{and} \tag{D.2}$$

$$c_{N,M} := \left(2\sup_{\alpha'\in\mathcal{A}_1}\left|\frac{1}{N}\sum_{n=1}^{N}\langle\eta_n,\Phi(u_n;\alpha')\rangle_{\mathcal{Y}}\right|\right)^2\,. \tag{D.3}$$

The inequality in (4.5) and the arithmetic-mean–geometric-mean inequality imply that

$$\lambda t \leq \mathscr{R}_N^\lambda(\widehat{\alpha};\mathcal{G}) \leq A_{N,M}^\lambda + \sqrt{c_{N,M}t} \leq A_{N,M}^\lambda + \frac{1}{2}\lambda^{-1}c_{N,M} + \frac{1}{2}\lambda t\,. \tag{D.4}$$

Subtracting $\lambda t/2$ from both sides and multiplying through by $2\lambda^{-1}$ yields

$$t \leq 2\lambda^{-1}A_{N,M}^\lambda + \lambda^{-2}c_{N,M}\,. \tag{D.5}$$

Substituting (D.5) back into the right-most side of (D.4) gives

$$\mathscr{R}_N^\lambda(\widehat{\alpha};\mathcal{G}) \leq 2A_{N,M}^\lambda + \lambda^{-1}c_{N,M}\,. \tag{D.6}$$

All of the above calculations hold with probability one. To complete our estimate of $\mathscr{R}_N^\lambda(\widehat{\alpha};\mathcal{G})$, it remains to upper bound the $A_{N,M}^\lambda$ (D.2) and $c_{N,M}$ (D.3) terms. We begin with the latter.

Lemmas D.4 and D.5 (with $t = 1$) and (A.7) show that

$$\sqrt{c_{N,M}} \leq \frac{4\|\eta\|_{\psi_1(\mathcal{Y})}\|\varphi\|_{L^\infty}}{\sqrt{N}} + 16e^{3/2}\|\eta\|_{\psi_1(\mathcal{Y})}\|\varphi\|_{L^\infty}\sqrt{\frac{\log(1/\delta)}{N}}$$

$$\leq 16e^{3/2}\|\eta\|_{\psi_1(\mathcal{Y})}\|\varphi\|_{L^\infty}\sqrt{\frac{6\log(2/\delta)}{N}} \tag{D.7}$$

with probability at least $1 - \delta$ if $N \geq \log(2/\delta) \geq \log(1/\delta)$. We used the inequalities $4 \leq 16e^{3/2}$, $\sqrt{a} + \sqrt{b} \leq \sqrt{2(a+b)}$, and $1 \leq 2\log(2/\delta)$ to get to the last line.

Continuing, we bound $A_{N,M}^\lambda$. It has several error contributions originating from two terms. The first term in (D.2) is $\mathscr{R}_N^\lambda(\alpha;\mathcal{G})$, where $\alpha\in\mathbb{R}^M$ is arbitrary. By Assumption 3.2, $\mathcal{G} = \rho + \mathcal{G}_{\mathcal{H}}$ so that

$$\mathscr{R}_N^\lambda(\alpha;\mathcal{G}) \leq \lambda\|\alpha\|_M^2 + 2\mathscr{R}_N(\alpha;\mathcal{G}_{\mathcal{H}}) + \frac{2}{N}\sum_{n=1}^{N}\|\rho(u_n)\|_{\mathcal{Y}}^2 \leq 2\mathscr{R}_N^\lambda(\alpha;\mathcal{G}_{\mathcal{H}}) + \frac{2}{N}\sum_{n=1}^{N}\|\rho(u_n)\|_{\mathcal{Y}}^2\,. \tag{D.8}$$

By Lem. 4.5, it holds with probability at least $1 - \delta$ that

$$\frac{2}{N}\sum_{n=1}^{N}\|\rho(u_n)\|_{\mathcal{Y}}^2 \leq 4\,\mathbb{E}_u\|\rho(u)\|_{\mathcal{Y}}^2 + \frac{9\|\rho\|_{L_\nu^\infty}^2\log(2/\delta)}{N}\,. \tag{D.9}$$

Next, we bound the first term on the right hand side in (D.8). Since $\mathcal{G}_{\mathcal{H}}\in\mathcal{H}$, there exists $\alpha_{\mathcal{H}}\in L_\mu^2(\Theta;\mathbb{R})$ such that

$$\mathcal{G}_{\mathcal{H}} = \mathbb{E}_{\theta\sim\mu}\big[\alpha_{\mathcal{H}}(\theta)\varphi(\,\cdot\,;\theta)\big] \quad\text{and}\quad \|\mathcal{G}_{\mathcal{H}}\|_{\mathcal{H}}^2 = \mathbb{E}_{\theta\sim\mu}|\alpha_{\mathcal{H}}(\theta)|^2\,.$$

With $\alpha_{\mathcal{H}}$ as in the above display, choose once and for all $\alpha \equiv \alpha^\star \in \mathbb{R}^M$ as in (4.7). By Lem. 4.3,

$$2\mathscr{R}_N^\lambda(\alpha^\star; \mathcal{G}_{\mathcal{H}}) \leq 162\lambda\|\mathcal{G}_{\mathcal{H}}\|_{\mathcal{H}}^2 \tag{D.10}$$

with probability at least $1 - \delta$ if $M \geq \lambda^{-1} \log(4/\delta)$. On the same event,

$$\|\alpha^\star\|_M^2 \leq \mathbb{E}_\theta |\alpha_{\mathcal{H}}(\theta)|^2 \left(1 + \frac{2\log(2/\delta)}{\lambda M} + \sqrt{\frac{2\log(2/\delta)}{\lambda M}}\right) \leq 5\,\mathbb{E}_\theta |\alpha_{\mathcal{H}}(\theta)|^2 = 5\|\mathcal{G}_{\mathcal{H}}\|_{\mathcal{H}}^2$$

by Lem. D.3. This fact and Lem. 4.6 (with $\alpha \equiv \alpha^\star$ as in (4.7)) shows that the second and final term in $A_{N,M}^\lambda$ (D.2) satisfies with probability at least $1 - \delta$ the upper bound

$$\frac{2}{N}\sum_{n=1}^N \langle -\eta_n, \Phi(u_n; \alpha^\star)\rangle_{\mathcal{Y}} \leq 40e^{3/2}\|\mathcal{G}_{\mathcal{H}}\|_{\mathcal{H}}\|\eta\|_{\psi_1(\mathcal{Y})}\|\varphi\|_{L^\infty}\sqrt{\frac{\log(2/\delta)}{N}}\,. \tag{D.11}$$

Combining the estimates (D.7), (D.8), (D.9), (D.10), and (D.11), recalling (D.6), and invoking the union bound, we deduce that if $N \geq \lambda^{-2}\log(2/\delta)$, then

$$\mathscr{R}_N^\lambda(\widehat{\alpha}; \mathcal{G}) \leq C_0\lambda + 8\,\mathbb{E}_{u\sim\nu}\|\rho(u)\|_{\mathcal{Y}}^2 + 18\|\rho\|_{L_\nu^\infty}^2\lambda^2$$

with probability at least $1 - 4\delta$, where

$$C_0 := 324\|\mathcal{G}_{\mathcal{H}}\|_{\mathcal{H}}^2 + 80e^{3/2}\|\eta\|_{\psi_1(\mathcal{Y})}\|\varphi\|_{L^\infty}\|\mathcal{G}_{\mathcal{H}}\|_{\mathcal{H}} + 1536e^3\|\eta\|_{\psi_1(\mathcal{Y})}^2\|\varphi\|_{L^\infty}^2$$
$$\leq (324 + 4)\|\mathcal{G}_{\mathcal{H}}\|_{\mathcal{H}}^2 + 1936e^3\|\eta\|_{\psi_1(\mathcal{Y})}^2\|\varphi\|_{L^\infty}^2\,.$$

In the last line, we used Young's inequality with $\varepsilon = 1/8$, that is, $ab \leq \varepsilon a^2/2 + b^2/(2\varepsilon)$ with $a = 80e^{3/2}\|\eta\|_{\psi_1(\mathcal{Y})}\|\varphi\|_{L^\infty}$ and $b = \|\mathcal{G}_{\mathcal{H}}\|_{\mathcal{H}}$. This proves the asserted upper bound. $\qquad\square$

### D.1.1  Proofs for subsection 4.1.1: Bounding the approximation error

Given a function $\alpha \in L_\mu^2(\Theta; \mathbb{R})$, we denote its cut-off at level $T > 0$ by

$$\theta \mapsto \alpha_{\leq T}(\theta) := \alpha(\theta)\mathbb{1}_{\{|\alpha(\theta)|\leq T\}}\,. \tag{D.12}$$

This subsection is devoted to the proof of Lem. 4.3, which is based on the following three lemmas.

**Lemma D.1.** *Suppose $\mathcal{G} \in \mathcal{H}$ belongs to the RKHS $\mathcal{H}$. Let $\alpha \in L_\mu^2(\Theta; \mathbb{R})$ be such that $\mathcal{G} = \mathbb{E}_\theta[\alpha(\theta)\varphi(\,\cdot\,; \theta)]$. Let $u_1, \ldots, u_N \sim \nu$ be iid samples and $\nu_N = \frac{1}{N}\sum_{n=1}^N \delta_{u_n}$ be the corresponding empirical measure. Then almost surely,*

$$\left\|\mathcal{G} - \mathbb{E}_\theta[\alpha_{\leq T}(\theta)\varphi(\,\cdot\,; \theta)]\right\|_{L_{\nu_N}^2}^2 \leq \frac{\|\varphi\|_{L^\infty}^2(\mathbb{E}_\theta |\alpha(\theta)|^2)^2}{T^2} \quad \textit{for all} \quad T > 0\,. \tag{D.13}$$

*Proof.* Fix $u \in \mathcal{X}$ and define $\alpha_{>T} := \alpha - \alpha_{\leq T}$. The claim follows from

$$\|\mathcal{G}(u) - \mathbb{E}_\theta[\alpha_{\leq T}(\theta)\varphi(u; \theta)]\|_{\mathcal{Y}}^2 = \|\mathbb{E}_\theta[\alpha_{>T}(\theta)\varphi(u; \theta)]\|_{\mathcal{Y}}^2 \leq \|\varphi\|_{L^\infty}^2(\mathbb{E}_\theta |\alpha_{>T}(\theta)|)^2$$

and the observation that $\mathbb{E}_\theta |\alpha_{>T}(\theta)| \leq \mathbb{E}_\theta |\alpha(\theta)|^2/T$. $\qquad\square$

The previous lemma controls the error incurred by truncating the coefficient function of elements in the RKHS. The next lemma provides a bound on sample average approximations of these truncations.

**Lemma D.2.** *Let $u_1, \ldots, u_N \sim \nu$ be iid samples and let $\nu_N = \frac{1}{N}\sum_{n=1}^N \delta_{u_n}$ denote the corresponding empirical measure. For $\alpha \in L_\mu^2(\Theta; \mathbb{R})$, let $Z = Z(\theta)$ be the $L_{\nu_N}^2(\mathcal{X}; \mathcal{Y})$-valued random variable defined for $\theta \sim \mu$ by*

$$Z = \alpha_{\leq T}(\theta)\varphi(\,\cdot\,; \theta)\,. \tag{D.14}$$

*If $Z_1, \ldots, Z_M$ are $M$ iid copies of $Z$, then it holds with probability at least $1 - \delta$ that*

$$\left\|\frac{1}{M}\sum_{m=1}^M Z_m - \mathbb{E}\,Z\right\|_{L_{\nu_N}^2}^2 \leq \frac{32T^2\|\varphi\|_{L^\infty}^2\log^2(2/\delta)}{M^2} + \frac{4\|\varphi\|_{L^\infty}^2\log(2/\delta)\,\mathbb{E}_\theta |\alpha(\theta)|^2}{M}\,. \tag{D.15}$$

*Proof.* By boundedness of $|\alpha_{\leq T}| \leq T$, we have the trivial uniform upper bound $\|Z_m\|_{L^2_{\nu_N}} \leq T\|\varphi\|_{L^\infty}$ for each $m$. The variance is bounded above as

$$\sigma^2 := \mathbb{E}\|Z - \mathbb{E}\,Z\|^2_{L^2_{\nu_N}} \leq \mathbb{E}\|Z\|^2_{L^2_{\nu_N}} \leq \|\varphi\|^2_{L^\infty}\,\mathbb{E}_\theta\,|\alpha(\theta)|^2\,.$$

By Lem. A.2 and Thm. A.1, it holds with probability at least $1 - \delta$ that

$$\left\|\frac{1}{M}\sum_{m=1}^{M} Z_m - \mathbb{E}\,Z\right\|_{L^2_{\nu_N}} \leq \frac{4T\|\varphi\|_{L^\infty}\log(2/\delta)}{M} + \sqrt{\frac{2\sigma^2\log(2/\delta)}{M}}\,.$$

Squaring both sides and substitution of the above bound on $\sigma^2$ yields the claimed estimate. $\qquad\square$

The third lemma below develops a high probability bound on the empirical approximation of the RKHS norm of truncated elements in the RKHS.

**Lemma D.3.** *Let $\alpha \in L^2_\mu(\Theta;\mathbb{R})$ and $\{\theta_m\} \sim \mu^{\otimes M}$. With probability at least $1 - \delta$, it holds that*

$$\frac{1}{M}\sum_{m=1}^{M}|\alpha_{\leq T}(\theta_m)|^2 \leq \mathbb{E}_\theta\,|\alpha(\theta)|^2 + \frac{4T^2\log(2/\delta)}{M} + \sqrt{\frac{2T^2\,\mathbb{E}_\theta\,|\alpha(\theta)|^2\log(2/\delta)}{M}}\,. \qquad \text{(D.16)}$$

*Proof.* We apply Bernstein's inequality (A.2) to the random variable $Z(\theta) := |\alpha_{\leq T}(\theta)|^2$ with $\theta \sim \mu$ and $M \in \mathbb{N}$ iid copies $Z_1, \ldots, Z_M$ of $Z$ defined by $Z_m = Z(\theta_m)$ for each $m$. We note that $|Z| \leq T^2$ by definition. The variance of $Z$ satisfies the upper bound

$$\sigma^2 := \mathbb{E}|Z - \mathbb{E}\,Z|^2 \leq \mathbb{E}|Z|^2 = \mathbb{E}_\theta\,|\alpha_{\leq T}(\theta)|^4 \leq T^2\,\mathbb{E}_\theta\,|\alpha(\theta)|^2\,.$$

It follows from Lem. A.2 and Thm. A.1 that

$$\frac{1}{m}\sum_{m=1}^{M} Z_m \leq \mathbb{E}\,Z + \frac{4T^2\log(2/\delta)}{M} + \sqrt{\frac{2\sigma^2\log(2/\delta)}{M}}$$

with probability at least $1 - \delta$. This in turn implies that

$$\frac{1}{M}\sum_{m=1}^{M}|\alpha_{\leq T}(\theta_m)|^2 \leq \mathbb{E}_\theta\,|\alpha(\theta)|^2] + \frac{4T^2\log(2/\delta)}{M} + \sqrt{\frac{2T^2\,\mathbb{E}\,|\alpha(\theta)|^2\log(2/\delta)}{M}}$$

with at least the same probability. This is the claim. $\qquad\square$

We are now in a position to prove Lem. 4.3.

*Proof of Lemma 4.3.* Write $T := T(\lambda)$ and $\alpha := \alpha_{\mathcal{H}}$. Next, define $\alpha_{\leq T} \in L^2_\mu(\Theta;\mathbb{R})$ by

$$\theta \mapsto \alpha_{\leq T}(\theta) := \alpha(\theta)\mathbb{1}_{\{|\alpha(\theta)|\leq T\}} = \begin{cases} \alpha(\theta), & \text{if } |\alpha(\theta)| \leq T\,, \\ 0, & \text{otherwise}\,. \end{cases}$$

We define $\alpha_{>T} := \alpha\mathbb{1}_{\{|\alpha|>T\}}$ similarly, so that $\alpha \equiv \alpha_{\leq T} + \alpha_{>T}$ holds true. Then for $\theta_1, \ldots, \theta_M$, we have $\alpha^\star \in \mathbb{R}^M$ given by $\alpha_m^\star = \alpha_{\leq T}(\theta_m)$ for each $m \in \{1, \ldots, M\}$. We claim that $\mathscr{R}_N^\lambda(\alpha^\star;\mathcal{G}) \leq (74\|\varphi\|^2_{L^\infty} + 7)\lambda\,\mathbb{E}_\theta\,|\alpha_{\mathcal{H}}(\theta)|^2$ with high probability, which implies the asserted bound (4.8). To see this, we make the error decomposition

$$\mathscr{R}_N^\lambda(\alpha^\star;\mathcal{G}) = \frac{1}{N}\sum_{n=1}^{N}\left\|\mathcal{G}(u_n) - \frac{1}{M}\sum_{m=1}^{M}\alpha_{\leq T}(\theta_m)\varphi(u_n;\theta_m)\right\|^2_{\mathcal{Y}} + \frac{\lambda}{M}\sum_{m=1}^{M}|\alpha_{\leq T}(\theta_m)|^2 \quad \text{(D.17)}$$

$$\leq \frac{2}{N}\sum_{n=1}^{N}\|\mathcal{G}(u_n) - \mathbb{E}_\theta[\alpha_{\leq T}(\theta)\varphi(u_n;\theta)]\|^2_{\mathcal{Y}} \qquad\qquad\qquad\qquad\text{(I)}$$

$$+ \frac{2}{N}\sum_{n=1}^{N}\left\|\frac{1}{M}\sum_{m=1}^{M}\alpha_{\leq T}(\theta_m)\varphi(u_n;\theta_m) - \mathbb{E}_\theta[\alpha_{\leq T}(\theta)\varphi(u_n;\theta)]\right\|^2_{\mathcal{Y}} \qquad\text{(II)}$$

$$+ \frac{\lambda}{M}\sum_{m=1}^{M}|\alpha_{\leq T}(\theta_m)|^2\,. \qquad\qquad\qquad\qquad\qquad\qquad\qquad\text{(III)}$$

Each of the three terms (I)–(III) is estimated as follows.

By Lem. D.1, we can bound

$$(\mathrm{I}) \le \frac{2\|\varphi\|_{L^\infty}^2 (\mathbb{E}_\theta |\alpha(\theta)|^2)^2}{T^2} = 2\lambda \|\varphi\|_{L^\infty}^2 \mathbb{E}_\theta |\alpha(\theta)|^2 \, .$$

Lem. D.2 delivers the bound

$$(\mathrm{II}) \le \frac{64 T^2 \|\varphi\|_{L^\infty}^2 \log(2/\delta)^2}{M^2} + \frac{8\|\varphi\|_{L^\infty}^2 \log(2/\delta) \, \mathbb{E}_\theta |\alpha(\theta)|^2}{M}$$

$$= \lambda \, \mathbb{E}_\theta |\alpha(\theta)|^2 \left( \frac{64 \|\varphi\|_{L^\infty}^2 \log(2/\delta)^2}{\lambda^2 M^2} + \frac{8\|\varphi\|_{L^\infty}^2 \log(2/\delta)}{\lambda M} \right)$$

with probability at least $1 - \delta$.

Last, Lem. D.3 yields

$$(\mathrm{III}) \le \lambda \, \mathbb{E} |\alpha(\theta)|^2 + \frac{4\lambda T^2 \log(2/\delta)}{M} + \lambda \sqrt{\frac{2 T^2 \, \mathbb{E} |\alpha(\theta)|^2 \log(2/\delta)}{M}}$$

$$= \lambda \, \mathbb{E}_\theta |\alpha(\theta)|^2 \left( 1 + \frac{4 \log(2/\delta)}{\lambda M} + \sqrt{\frac{2 \log(2/\delta)}{\lambda M}} \right)$$

with probability at least $1 - \delta$.

Combining the three estimates, it follows that if

$$\frac{\log(2/\delta)}{\lambda M} \le 1 \, ,$$

then

$$\mathscr{R}_N^\lambda(\alpha^\star; \mathcal{G}_\mathcal{H}) = \mathscr{R}_N^\lambda(\alpha^\star; \mathcal{G}) \le (74 \|\varphi\|_{L^\infty}^2 + 7)\lambda \, \mathbb{E}_\theta |\alpha_\mathcal{H}(\theta)|^2$$

with probability at least $1 - 2\delta$. We used the fact that $\sqrt{2} \le 2$. This is the claimed upper bound. $\square$

### D.1.2 Proof for subsection 4.1.2: Bounding the misspecification error

Recall that $\rho \in L_\nu^\infty$ under Assumption 3.2. We now prove Lem. 4.5.

*Proof of Lemma 4.5.* Let $Z_1 = \|\rho(u_1)\|_\mathcal{Y}^2$, which is uncentered. Almost surely, $Z_1 \le \|\rho\|_{L_\nu^\infty}^2$ and

$$\mathbb{E}|Z_1 - \mathbb{E} Z_1|^2 \le \mathbb{E} Z_1^2 = \mathbb{E}_{u \sim \nu} \|\rho(u)\|_\mathcal{Y}^4 \le \|\rho\|_{L_\nu^\infty}^2 \, \mathbb{E}_{u \sim \nu} \|\rho(u)\|_\mathcal{Y}^2 \, .$$

Thus with probability at least $1 - \delta$, Cor. A.2 and Thm. A.1 provide the bound

$$\frac{1}{N} \sum_{n=1}^N \|\rho(u_n)\|_\mathcal{Y}^2 \le \mathbb{E}_u \|\rho(u)\|_\mathcal{Y}^2 + \frac{4\|\rho\|_{L_\nu^\infty}^2 \log(2/\delta)}{N} + \sqrt{\frac{2 \, \mathbb{E}_u \|\rho(u)\|_\mathcal{Y}^2 \|\rho\|_{L_\nu^\infty}^2 \log(2/\delta)}{N}}$$

$$\le 2 \, \mathbb{E}_u \|\rho(u)\|_\mathcal{Y}^2 + \frac{\frac{9}{2}\|\rho\|_{L_\nu^\infty}^2 \log(2/\delta)}{N} \, .$$

To get the last inequality, we used the arithmetic-mean–geometric-mean inequality $\sqrt{ab} \le (a + b)/2$ to obtain

$$\sqrt{\left( 2 \, \mathbb{E}_u \|\rho(u)\|_\mathcal{Y}^2 \right) \left( \|\rho\|_{L_\nu^\infty}^2 \log(2/\delta)/N \right)} \le \mathbb{E}_u \|\rho(u)\|_\mathcal{Y}^2 + \frac{\frac{1}{2}\|\rho\|_{L_\nu^\infty}^2 \log(2/\delta)}{N} \, .$$

Multiplying the penultimate chain of inequalities through by two completes the proof. $\square$

### D.1.3  Proofs for subsection 4.1.3: Bounding the noise error

This subsection provides proofs for the lemmas used to control the error stemming from iid noise corrupting the output data as in Assumption 3.3. The estimates themselves could be improved by using (A.10) instead of (A.11) and by tracking the noise variance $\mathbb{E}\|\eta\|_{\mathcal{Y}}^2$ instead of bounding it above by $4\|\eta\|_{\psi_1(\mathcal{Y})}^2$. This would be relevant in settings where the noise is small or tends to zero with the sample size. We defer such considerations to future work.

*Proof of Lemma 4.6.* Define $Z_n(\alpha) := \langle -\eta_n, \Phi(u_n; \alpha)\rangle_{\mathcal{Y}}$ for each $n$. Conditioned on $\{\theta_m\}$, it holds that $Z_n$ is an iid copy of $Z_1$. By the assumption $\mathbb{E}[\eta_1 \mid u_1] = 0$, we have

$$
\begin{aligned}
\mathbb{E}\, Z_1(\alpha) &= \mathbb{E}_{(u_1,\eta_1)}[\langle -\eta_1, \Phi(u_1;\alpha)\rangle_{\mathcal{Y}}] \\
&= \mathbb{E}_{u_1 \sim \nu}[\mathbb{E}[\langle -\eta_1, \Phi(u_1;\alpha)\rangle_{\mathcal{Y}} \mid u_1]] \\
&= \mathbb{E}_{u_1 \sim \nu}[\langle -\mathbb{E}[\eta_1 \mid u_1], \Phi(u_1;\alpha)\rangle_{\mathcal{Y}}] \\
&= 0\,.
\end{aligned}
\tag{D.18}
$$

Next,

$$
|Z_1(\alpha)| \le \|\eta_1\|_{\mathcal{Y}}\|\Phi(u_1;\alpha)\|_{\mathcal{Y}} \le \|\eta_1\|_{\mathcal{Y}}\|\alpha\|_M\|\varphi\|_{L^\infty}
$$

by two applications of the Cauchy–Schwarz inequality, one in $\mathcal{Y}$ and the other in $\mathbb{R}^M$. We deduce that $\|Z_1(\alpha)\|_{\psi_1} \le \|\eta_1\|_{\psi_1(\mathcal{Y})}\|\alpha\|_M\|\varphi\|_{L^\infty}$, conditioned on $\{\theta_m\}$. Prop. A.4, Thm. A.1 (Bernstein's inequality), and a similar argument to that in the proof of Cor. A.5 deliver the asserted bound. □

The next two lemmas are used in the proof of Prop. 4.1 to control the third and final term in (4.5).

**Lemma D.4** (Linear empirical process: Concentration)**.** *Fix $t > 0$ and $\delta \in (0,1)$. Define*

$$
Z_t := \sup_{\alpha \in \mathcal{A}_t}\left|\frac{1}{N}\sum_{n=1}^N \langle \eta_n, \Phi(u_n;\alpha)\rangle_{\mathcal{Y}}\right|, \quad \text{where} \quad \mathcal{A}_t := \left\{\alpha \in \mathbb{R}^M \,\middle|\, \|\alpha\|_M^2 \le t\right\}.
\tag{D.19}
$$

*If $N \ge \log(1/\delta)$, then conditioned on the realizations $\{\theta_m\}$ in the family $\Phi$ it holds that*

$$
Z_t \le \mathbb{E}_{\{(u_n,\eta_n)\}}[Z_t] + 8e^{3/2}\|\eta_1\|_{\psi_1(\mathcal{Y})}\|\varphi\|_{L^\infty}\sqrt{\frac{\log(1/\delta)}{N}}\sqrt{t}
\tag{D.20}
$$

*with probability at least $1 - \delta$.*

*Proof.* For any $\alpha \in \mathcal{A}_t$, we compute

$$
\begin{aligned}
|\langle \eta_1, \Phi(u_1;\alpha)\rangle_{\mathcal{Y}}| &= \left|\frac{1}{M}\sum_{m=1}^M \alpha_m \langle \eta_1, \varphi(u_1;\theta_m)\rangle_{\mathcal{Y}}\right| \\
&\le \left(\frac{1}{M}\sum_{m=1}^M |\alpha_m|^2\right)^{1/2}\left(\frac{1}{M}\sum_{m=1}^M \langle \eta_1, \varphi(u_1;\theta_m)\rangle_{\mathcal{Y}}^2\right)^{1/2} \\
&\le \sqrt{t}\left(\|\eta_1\|_{\mathcal{Y}}^2 \frac{1}{M}\sum_{m=1}^M \|\varphi(u_1;\theta_m)\|_{\mathcal{Y}}^2\right)^{1/2}\,.
\end{aligned}
$$

We used the Cauchy–Schwarz inequality twice. By the boundedness of $\varphi$, the above display gives

$$
\big\|\langle \eta_1, \Phi(u_1;\,\cdot\,)\rangle\big\|_{\psi_1(C(\mathcal{A}_t;\mathbb{R}))} = \left\|\sup_{\alpha \in \mathcal{A}_t}|\langle \eta_1, \Phi(u_1;\alpha)\rangle_{\mathcal{Y}}|\right\|_{\psi_1} \le \|\eta_1\|_{\psi_1(\mathcal{Y})}\|\varphi\|_{L^\infty}\sqrt{t}\,.
$$

The iid random variables $\langle \eta_n, \Phi(u_n;\,\cdot\,)\rangle_{\mathcal{Y}}\colon \alpha \mapsto \langle \eta_n, \Phi(u_n;\alpha)\rangle_{\mathcal{Y}}$ (conditional on $\{\theta_m\}$) are linear and hence continuous. Application of (A.11) in Cor. A.5 to $\langle \eta_n, \Phi(u_n;\,\cdot\,)\rangle_{\mathcal{Y}}$ taking value in the separable Banach space $C(\mathcal{A}_t;\mathbb{R})$ of continuous functions from compact set $\mathcal{A}_t \subset \mathbb{R}^M$ into $\mathbb{R}$, equipped with the supremum norm, completes the proof. □

The previous lemma gives a concentration bound for the linear empirical process and the next lemma estimates its expectation.

**Lemma D.5** (Linear empirical process: Expectation). *Fix $t > 0$. Define $\mathcal{A}_t$ as in Lem. D.4. Then*

$$\mathbb{E}_{\{(u_n,\eta_n)\}} \sup_{\alpha \in \mathcal{A}_t} \left| \frac{1}{N} \sum_{n=1}^{N} \langle \eta_n, \Phi(u_n; \alpha) \rangle_{\mathcal{Y}} \right| \leq \frac{\|\eta_1\|_{L^2_{\mathbb{P}}(\Omega; \mathcal{Y})} \|\varphi\|_{L^\infty}}{\sqrt{N}} \sqrt{t}. \tag{D.21}$$

*Proof.* For any $\alpha \in \mathcal{A}_t$, the Cauchy–Schwarz inequality in $\mathbb{R}^M$ delivers the bound

$$\left| \frac{1}{N} \sum_{n=1}^{N} \langle \eta_n, \Phi(u_n; \alpha) \rangle_{\mathcal{Y}} \right| = \left| \frac{1}{M} \sum_{m=1}^{M} \alpha_m \frac{1}{N} \sum_{n=1}^{N} \langle \eta_n, \varphi(u_n; \theta_m) \rangle_{\mathcal{Y}} \right|$$

$$\leq \left( \frac{1}{M} \sum_{m=1}^{M} |\alpha_m|^2 \right)^{1/2} \left( \frac{1}{M} \sum_{m=1}^{M} \left[ \frac{1}{N} \sum_{n=1}^{N} \langle \eta_n, \varphi(u_n; \theta_m) \rangle_{\mathcal{Y}} \right]^2 \right)^{1/2}.$$

Let the left hand side of (D.21) be denoted by $\Xi_t$. We next note that

$$\mathbb{E}_{\{(u_n,\eta_n)\}} \left[ \langle \eta_n, \varphi(u_n; \theta_m) \rangle_{\mathcal{Y}} \right] = \mathbb{E}_{u_n \sim \nu} \left[ \mathbb{E} \left[ \langle \eta_n, \varphi(u_n; \theta_m) \rangle_{\mathcal{Y}} \mid u_n \right] \right]$$

$$= \mathbb{E}_{u_n \sim \nu} [ \langle \mathbb{E}[\eta_n \mid u_n], \varphi(u_n; \theta_m) \rangle_{\mathcal{Y}} ]$$

$$= 0.$$

Using the independence of $(u_n, \eta_n)$ and $(u_{n'}, \eta_{n'})$ for any two indices $n \neq n'$, together with the above observation, we thus obtain

$$\mathbb{E}_{\{(u_n,\eta_n)\}} \left[ \langle \eta_n, \varphi(u_n; \theta_m) \rangle_{\mathcal{Y}} \langle \eta_{n'}, \varphi(u_{n'}; \theta_m) \rangle_{\mathcal{Y}} \right]$$

$$= \mathbb{E}_{(u_n,\eta_n)} \left[ \langle \eta_n, \varphi(u_n; \theta_m) \rangle_{\mathcal{Y}} \right] \mathbb{E}_{(u_{n'},\eta_{n'})} \left[ \langle \eta_{n'}, \varphi(u_{n'}; \theta_m) \rangle_{\mathcal{Y}} \right]$$

$$= 0.$$

This implies that

$$\Xi_t \leq \frac{\sqrt{t}}{N} \mathbb{E}_{\{(u_n,\eta_n)\}} \sqrt{\frac{1}{M} \sum_{m=1}^{M} \sum_{n,n'=1}^{N} \langle \eta_n, \varphi(u_n; \theta_m) \rangle_{\mathcal{Y}} \langle \eta_{n'}, \varphi(u_{n'}; \theta_m) \rangle_{\mathcal{Y}}}$$

$$\leq \frac{\sqrt{t}}{N} \sqrt{\frac{1}{M} \sum_{m=1}^{M} \sum_{n,n'=1}^{N} \mathbb{E}_{\{(u_n,\eta_n)\}} \left[ \langle \eta_n, \varphi(u_n; \theta_m) \rangle_{\mathcal{Y}} \langle \eta_{n'}, \varphi(u_{n'}; \theta_m) \rangle_{\mathcal{Y}} \right]}$$

$$= \frac{\sqrt{t}}{\sqrt{N}} \sqrt{\frac{1}{M} \sum_{m=1}^{M} \mathbb{E}_{(u_1,\eta_1)} \langle \eta_1, \varphi(u_1; \theta_m) \rangle_{\mathcal{Y}}^2}$$

$$\leq \frac{\sqrt{t}}{\sqrt{N}} \|\eta_1\|_{L^2_{\mathbb{P}}(\Omega; \mathcal{Y})} \|\varphi\|_{L^\infty}.$$

We used Jensen's inequality in the second line, independence and the zero-mean property of the summands in the third line, and the Cauchy–Schwarz inequality in $\mathcal{Y}$ in the final line. □

## D.2 Proofs for subsection 4.2: Bounding the generalization gap

This subsection upper bounds the generalization gap with suprema techniques. We begin with the following empirical process concentration inequality. It gives uniform control on the difference between the empirical and population risk functionals. The process, as a function of its index $\alpha$, is quadratic because the RFM $\Phi(\cdot; \alpha)$ is linear in $\alpha$.

**Lemma D.6** (Quadratic empirical process: Concentration). *Fix $t > 0$ and $\delta \in (0,1)$. Define*

$$Z_t := \sup_{\alpha \in \mathcal{A}_t} \left| \mathscr{R}_N(\alpha; \mathcal{G}) - \mathscr{R}(\alpha; \mathcal{G}) \right| \tag{D.22}$$

$$= \sup_{\alpha \in \mathcal{A}_t} \left| \frac{1}{N} \sum_{n=1}^{N} \|\mathcal{G}(u_n) - \Phi(u_n; \alpha)\|_{\mathcal{Y}}^2 - \mathbb{E}_{u \sim \nu} \|\mathcal{G}(u) - \Phi(u; \alpha)\|_{\mathcal{Y}}^2 \right|, \tag{D.23}$$

*where*

$$\mathcal{A}_t := \left\{ \alpha \in \mathbb{R}^M \,\middle|\, \|\alpha\|_M^2 \leq t \right\}. \tag{D.24}$$

*If $N \geq \log(1/\delta)$, then conditioned on the realizations $\{\theta_m\}$ in the family $\Phi$, it holds that*

$$Z_t \leq \mathbb{E}_{\{u_n\}}[Z_t] + 32 e^{3/2} (\|\mathcal{G}\|_{L_\nu^\infty}^2 + \|\varphi\|_{L^\infty}^2 t) \sqrt{\frac{\log(1/\delta)}{N}} \tag{D.25}$$

*with probability at least $1 - \delta$.*

*Proof.* For any $\alpha \in \mathcal{A}_t$ and $n \in \{1, \dots, N\}$, let

$$X_n(t, \alpha) := \|\mathcal{G}(u_n) - \Phi(u_n; \alpha)\|_{\mathcal{Y}}^2 - \mathbb{E}_{u \sim \nu} \|\mathcal{G}(u) - \Phi(u; \alpha)\|_{\mathcal{Y}}^2.$$

We compute

$$|X_1(t, \alpha)| \leq 2\|\mathcal{G}(u_1)\|_{\mathcal{Y}}^2 + 2\,\mathbb{E}_{u \sim \nu} \|\mathcal{G}(u)\|_{\mathcal{Y}}^2 + 2\|\Phi(u_1; \alpha)\|_{\mathcal{Y}}^2 + 2\,\mathbb{E}_{u \sim \nu} \|\Phi(u; \alpha)\|_{\mathcal{Y}}^2$$
$$\leq 4\|\mathcal{G}\|_{L_\nu^\infty}^2 + 4\|\varphi\|_{L^\infty}^2 t.$$

We used the fact that for any $u \in \mathcal{X}$ $\nu$-almost surely, $\|\Phi(u; \alpha)\|_{\mathcal{Y}}^2 \leq t\|\varphi\|_{L^\infty}^2$ on the set $\mathcal{A}_t$ (by the Cauchy–Schwarz inequality). This implies that

$$\left\| X_1(t, \cdot) \right\|_{\psi_1(C(\mathcal{A}_t; \mathbb{R}))} = \left\| \sup_{\alpha \in \mathcal{A}_t} |X_1(t, \alpha)| \right\|_{\psi_1} \leq 4\|\mathcal{G}\|_{L_\nu^\infty}^2 + 4\|\varphi\|_{L^\infty}^2 t.$$

The $X_n(t, \cdot)$ do indeed belong to $C(\mathcal{A}_t; \mathbb{R})$ almost surely, as they can be written as a sum of affine and quadratic forms on $\mathbb{R}^M$ in the $\alpha$ variable. Application of (A.11) in Cor. A.5 (taking the separable Banach space to be $C(\mathcal{A}_t; \mathbb{R})$ equipped with the supremum norm) completes the proof. $\qquad\square$

Since the supremum concentrates around its mean, it remains to show that its mean is small as a function of the sample size. The next lemma does this with Rademacher symmetrization.

**Lemma D.7** (Quadratic empirical process: Expectation)**.** *Fix $t > 0$. Define $Z_t$ as in Lem. D.6. Conditioned on the realizations $\{\theta_m\}$ in the family $\Phi$, it holds that*

$$\mathbb{E}_{\{u_n\}}[Z_t] \leq \frac{4\|\mathcal{G}\|_{L_\nu^\infty}^2}{\sqrt{N}} + \frac{4\|\varphi\|_{L^\infty}^2}{\sqrt{N}} t. \tag{D.26}$$

*Proof.* By Giné–Zinn symmetrization [see, e.g., 46, Sect. 4.2, Prop. 4.11, pp. 107–108],

$$\mathbb{E}_{\{u_n\}}[Z_t] \leq 2\,\mathbb{E} \sup_{\alpha \in \mathcal{A}_t} \left| \frac{1}{N} \sum_{n=1}^N \varepsilon_n \|\mathcal{G}(u_n) - \Phi(u_n; \alpha)\|_{\mathcal{Y}}^2 \right|, \quad \text{where} \quad \varepsilon_n \overset{\text{iid}}{\sim} \mathrm{Unif}\big(\{+1, -1\}\big),$$

because the original summands (conditioned on $\{\theta_m\}$) are independent. The expectation on the right is interpreted as the conditional expectation given $\{\theta_m\}$ (i.e., $\mathbb{E}_{\{u_n\},\{\varepsilon_n\}}$ over the data and Rademacher variables only). The right hand side is the Rademacher complexity of the RF model class composed with the square loss. Expanding the square, it is bounded above by

$$2\,\mathbb{E}_{\{u_n\},\{\varepsilon_n\}} \left| \frac{1}{N} \sum_{n=1}^N \varepsilon_n \|\mathcal{G}(u_n)\|_{\mathcal{Y}}^2 \right| \tag{I}$$

$$+ 4\,\mathbb{E}_{\{u_n\},\{\varepsilon_n\}} \sup_{\alpha \in \mathcal{A}_t} \left| \frac{1}{N} \sum_{n=1}^N \varepsilon_n \langle \mathcal{G}(u_n), \Phi(u_n; \alpha) \rangle_{\mathcal{Y}} \right| \tag{II}$$

$$+ 2\,\mathbb{E}_{\{u_n\},\{\varepsilon_n\}} \sup_{\alpha \in \mathcal{A}_t} \left| \frac{1}{N} \sum_{n=1}^N \varepsilon_n \|\Phi(u_n; \alpha)\|_{\mathcal{Y}}^2 \right|. \tag{III}$$

We now estimate each term. The first term (I) satisfies the standard Monte Carlo bound

$$\text{(I)} \leq 2 \left( \mathbb{E} \left| \frac{1}{N} \sum_{n=1}^N \varepsilon_n \|\mathcal{G}(u_n)\|_{\mathcal{Y}}^2 \right|^2 \right)^{1/2} = \frac{2}{\sqrt{N}} \left( \frac{1}{N} \sum_{n=1}^N \mathbb{E}\|\mathcal{G}(u_n)\|_{\mathcal{Y}}^4 \right)^{1/2} \leq \frac{2\|\mathcal{G}\|_{L_\nu^\infty}^2}{\sqrt{N}}.$$

For the second term (II), we begin by estimating the empirical average on the set $\mathcal{A}_t$ as

$$\left| \frac{1}{N} \sum_{n=1}^{N} \varepsilon_n \langle \mathcal{G}(u_n), \Phi(u_n; \alpha) \rangle_{\mathcal{Y}} \right| = \left| \frac{1}{M} \sum_{m=1}^{M} \alpha_m \left( \frac{1}{N} \sum_{n=1}^{N} \varepsilon_n \langle \mathcal{G}(u_n), \varphi(u_n; \theta_m) \rangle_{\mathcal{Y}} \right) \right|$$

$$\leq \sqrt{t} \left( \frac{1}{M} \sum_{m=1}^{M} \left| \frac{1}{N} \sum_{n=1}^{N} \varepsilon_n \langle \mathcal{G}(u_n), \varphi(u_n; \theta_m) \rangle_{\mathcal{Y}} \right|^2 \right)^{1/2}$$

by the Cauchy–Schwarz inequality in $\mathbb{R}^M$. We deduce by Jensen's inequality and independence that

$$\text{(II)} \leq \frac{4\sqrt{t}}{N} \left( \frac{1}{M} \sum_{m=1}^{M} \sum_{n,n'=1}^{N} \mathbb{E}[\varepsilon_n \varepsilon_{n'}] \, \mathbb{E}_{\{u_n\}} \left[ \langle \mathcal{G}(u_n), \varphi(u_n; \theta_m) \rangle_{\mathcal{Y}} \langle \mathcal{G}(u_{n'}), \varphi(u_{n'}; \theta_m) \rangle_{\mathcal{Y}} \right] \right)^{1/2}$$

$$= \frac{4\sqrt{t}}{N} \left( \frac{1}{M} \sum_{m=1}^{M} \sum_{n=1}^{N} \mathbb{E}_{u \sim \nu} \langle \mathcal{G}(u), \varphi(u; \theta_m) \rangle_{\mathcal{Y}}^2 \right)^{1/2}.$$

A final application of the Cauchy–Schwarz inequality in $\mathcal{Y}$ in the last line shows that the second term (II) is bounded above by $4\sqrt{t} \|\mathcal{G}\|_{L_\nu^\infty} \|\varphi\|_{L^\infty} / \sqrt{N}$. By Young's inequality $ab \leq a^2/2 + b^2/2$, we further bound

$$\frac{4\|\mathcal{G}\|_{L_\nu^\infty} \|\varphi\|_{L^\infty} \sqrt{t}}{\sqrt{N}} = \left( \frac{2\|\mathcal{G}\|_{L_\nu^\infty}}{N^{1/4}} \right) \left( \frac{2\|\varphi\|_{L^\infty} \sqrt{t}}{N^{1/4}} \right) \leq \frac{2\|\mathcal{G}\|_{L_\nu^\infty}^2}{\sqrt{N}} + \frac{2\|\varphi\|_{L^\infty}^2 t}{\sqrt{N}}.$$

The third term (III) is estimated in a similar manner. Expanding the empirical average on $\mathcal{A}_t$ yields

$$\left| \frac{1}{N} \sum_{n=1}^{N} \varepsilon_n \|\Phi(u_n; \alpha)\|_{\mathcal{Y}}^2 \right| = \left| \frac{1}{M} \sum_{m=1}^{M} \alpha_m \left( \frac{1}{M} \sum_{m'=1}^{M} \alpha_{m'} \beta_{m,m'}^{(N)} \right) \right|, \quad \text{where}$$

$$\beta_{m,m'}^{(N)} = \frac{1}{N} \sum_{n=1}^{N} \varepsilon_n \langle \varphi(u_n; \theta_m), \varphi(u_n; \theta_{m'}) \rangle_{\mathcal{Y}}.$$

The first equality in the above display satisfies the upper bound

$$\sqrt{t} \left( \frac{1}{M} \sum_{m=1}^{M} \left| \frac{1}{M} \sum_{m'=1}^{M} \alpha_{m'} \beta_{m,m'}^{(N)} \right|^2 \right)^{1/2} \leq \sqrt{t} \left( \frac{1}{M} \sum_{m=1}^{M} t \left[ \frac{1}{M} \sum_{m'=1}^{M} |\beta_{m,m'}^{(N)}|^2 \right] \right)^{1/2}$$

$$= \frac{t}{N} \sqrt{ \frac{1}{M^2} \sum_{m,m'=1}^{M} \left| \sum_{n=1}^{N} \varepsilon_n \langle \varphi(u_n; \theta_m), \varphi(u_n; \theta_{m'}) \rangle_{\mathcal{Y}} \right|^2 }$$

by two applications of the Cauchy–Schwarz inequality in $\mathbb{R}^M$. Finally, we deduce that

$$\text{(III)} \leq \frac{2t}{N} \sqrt{ \frac{1}{M^2} \sum_{m,m'=1}^{M} \sum_{n=1}^{N} \mathbb{E}_{\{u_n\}} \langle \varphi(u_n; \theta_m), \varphi(u_n; \theta_{m'}) \rangle_{\mathcal{Y}}^2 }$$

$$\leq \frac{2t}{\sqrt{N}} \sqrt{ \frac{1}{M^2} \sum_{m,m'=1}^{M} \mathbb{E}_u \left[ \|\varphi(u; \theta_m)\|_{\mathcal{Y}}^2 \|\varphi(u; \theta_{m'})\|_{\mathcal{Y}}^2 \right] }$$

$$\leq \frac{2t\|\varphi\|_{L^\infty}^2}{\sqrt{N}}$$

by Jensen's inequality, the fact that $\mathbb{E}[\varepsilon_n \varepsilon_{n'}] = \delta_{n,n'}$, and the Cauchy–Schwarz inequality in $\mathcal{Y}$. Combining the three estimates completes the proof. $\square$

The proof of the main generalization gap bound (4.12) is now immediate.

*Proof of Proposition 4.7.* Lem. D.6 and D.7 (applied with $t = \beta$) show that

$$\mathcal{E}_\beta(\{u_n\}, \{\theta_m\}) \leq \frac{4(\|\mathcal{G}\|_{L_\nu^\infty}^2 + \|\varphi\|_{L^\infty}^2 \beta)}{\sqrt{N}} + 32e^{3/2}(\|\mathcal{G}\|_{L_\nu^\infty}^2 + \|\varphi\|_{L^\infty}^2 \beta)\sqrt{\frac{\log(1/\delta)}{N}} \quad \text{(D.27)}$$

with conditional probability (over $\{\theta_m\}$) at least $1 - \delta$ if $N \geq \log(1/\delta)$. Since $\delta$ does not depend on $\{\theta_m\}$, we deduce by the tower rule of conditional expectation that the event implied by (D.27) has $\mathbb{P}$-probability at least $1 - \delta$ as well. Using the inequalities $4 \leq 32e^{3/2}$ and $\sqrt{a} + \sqrt{b} \leq \sqrt{2(a+b)}$ shows that the expression in (D.27) is bounded above by

$$32e^{3/2}(\|\mathcal{G}\|_{L_\nu^\infty}^2 + \|\varphi\|_{L^\infty}^2 \beta)\sqrt{\frac{2(1 + \log(1/\delta))}{N}}\,.$$

Application of the inequality $1 \leq 2\log(2/\delta)$, valid for $\delta \in (0, 1)$, implies (4.12) as asserted. $\qquad\square$

# E   Numerical experiment details

In this appendix, we detail the setup of the numerical experiment from Sect. 5 and provide additional visualization of the function-valued RFM's discretization-independence in Figure 3. All code used to produce the numerical results and figures in this paper are available at

https://github.com/nickhnelsen/error-bounds-for-vvRF.

A RFM with $M$ features is trained on $N$ input-output pairs $\{(u_n, \mathcal{G}(u_n))\}_{n=1}^N$ according to the vector-valued RF-RR algorithm. The ground truth map $\mathcal{G}: L^2(\mathbb{T}; \mathbb{R}) \to L^2(\mathbb{T}; \mathbb{R})$ is a nonlinear operator defined by $u^{(0)}(\cdot) \mapsto u(\cdot, 1)$, where $u = \{u(x, t)\}_{x,t}$ solves the partial differential equation

$$\frac{\partial u}{\partial t} + \frac{\partial}{\partial x}\left(\frac{u^2}{2}\right) = 10^{-1}\frac{\partial^2 u}{\partial x^2}\,, \quad (x, t) \in \mathbb{T} \times (0, \infty)\,,$$

with initial condition $u(\cdot, 0) = u^{(0)} \in L^2(\mathbb{T}; \mathbb{R})$. Here, $\mathbb{T} \simeq (0, 2\pi)_{\text{per}}$ is the 1D torus which comes with periodic boundary conditions. The initial conditions $u_n \sim \nu$ are sampled iid from a centered Matérn-like Gaussian process according to [23, Sect. 6.3, p. 32].

The random features are defined in a similar way to the Fourier Space RFs in [32, Sect. 3.1, p. 15]:

$$\varphi(u^{(0)}; \theta) = 2.6 \cdot \text{ELU}\big(\mathcal{F}^{-1}\{1_{(|k| \leq k_{\max})}\chi_k \cdot (\mathcal{F}u^{(0)})_k \cdot (\mathcal{F}\theta)_k\}_{k \in \mathbb{Z}}\big) \quad \text{and} \quad \theta \sim \mu\,,$$

where $\mu$ is also a centered Matérn Gaussian measure with covariance operator $1.8^2(-\frac{d^2}{dx^2} + 15^2\,\text{Id})^{-3}$. In the above display, $\mathcal{F}$ maps a function to its Fourier series coefficients, and $\mathcal{F}^{-1}$ expresses a Fourier coefficient sequence as a function expanded in the Fourier basis. The filter $\chi$ is given by [32, Eqn. 3.6, p. 15] with $\delta = 0.32$ and $\beta = 0.1$. We take $k_{\max} = 64$. The feature map $\varphi$ lifts the notion of hidden neuron in neural network architectures to function space.

In Figures 2 and 3, the quantity represented on the vertical axis is an empirical approximation to the relative Bochner squared error:

$$\frac{\frac{1}{N'}\sum_{n=1}^{N'}\|\mathcal{G}(u_n') - \Phi(u_n'; \widehat{\alpha}, \{\theta_m\})\|_{L^2}^2}{\frac{1}{N'}\sum_{n=1}^{N'}\|\mathcal{G}(u_n')\|_{L^2}^2} \approx \frac{\mathbb{E}_{u \sim \nu}\|\mathcal{G}(u) - \Phi(u; \widehat{\alpha}, \{\theta_m\})\|_{L^2}^2}{\mathbb{E}_{u \sim \nu}\|\mathcal{G}(u)\|_{L^2}^2} = \frac{\mathscr{R}(\widehat{\alpha}; \mathcal{G})}{\mathscr{R}(0; \mathcal{G})}\,. \quad \text{(E.1)}$$

In (E.1), $N' = 500$ is the size of the test set $\{(u_n', \mathcal{G}(u_n'))\}_{n=1}^{N'}$, where $\{u_n'\}_{n=1}^{N'} \sim \nu^{\otimes N'}$ is disjoint from the training input set $\{u_n\}_{n=1}^N$. The input and output spaces are discretized on the same $p$-point equally spaced grid in $(0, 2\pi)$. Thus, the discretized version of any input or output function belonging to $\mathcal{X} = \mathcal{Y} = L^2(\mathbb{T}; \mathbb{R})$ may be identified with an element of $\mathbb{R}^p$. In Figures 2a and 3a, the regularization factor is chosen as $\lambda = 7 \cdot 10^{-4}/M$ and as $\lambda = 3 \cdot 10^{-6}/\sqrt{N}$ in Figures 2b and 3b.

A priori, it is not clear whether this operator learning benchmark satisfies our theoretical assumptions because we cannot verify that the Burgers' solution operator belongs to the RKHS of $(\varphi, \mu)$ (or the range of some power of the RKHS kernel integral operator). At a more technical level, the feature map $\varphi$ uses an unbounded activation function (ELU, the exponential linear unit), while our theory is only developed for bounded RFs (Assumption 3.1). Nevertheless, the empirically obtained parameter and sample complexity in Figure 2 reasonably fit the main result of our well-specified theory (Theorem 3.7).

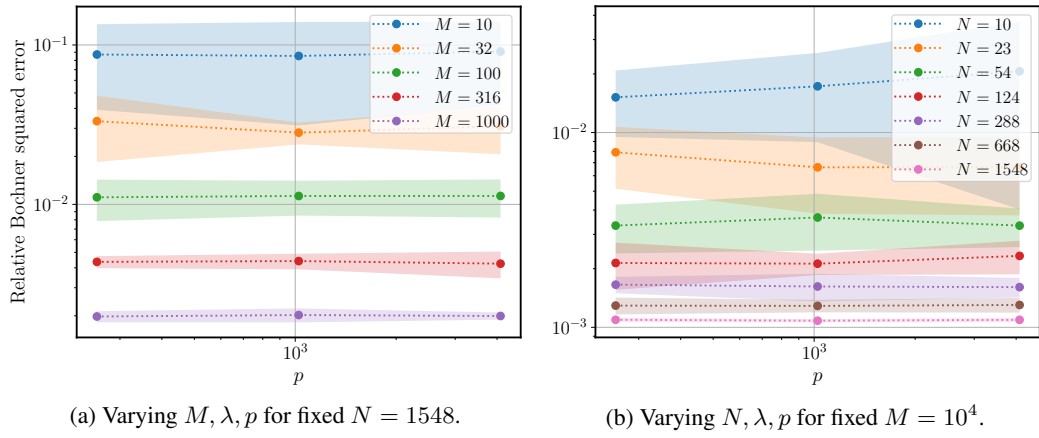

(a) Varying $M, \lambda, p$ for fixed $N = 1548$.   (b) Varying $N, \lambda, p$ for fixed $M = 10^4$.

Figure 3: Squared test error—which empirically approximates the population risk $\mathscr{R}(\widehat{\alpha}; \mathcal{G})$—versus discretized output space dimension $p$, where $\mathcal{G}$ is the Burgers' equation solution operator (**SM** E).

