# OpenReview forum: "Error Bounds for Learning with Vector-Valued Random Features"
_NeurIPS.cc/2023/Conference — NeurIPS 2023 spotlight_

### Official Review · Reviewer_HQhU · 2023-06-20

**Soundness:** 3 good
**Presentation:** 3 good
**Contribution:** 3 good
**Rating:** 6
**Confidence:** 4

**Summary:**

This paper presents error analysis for learning with vector-valued random features, where the output takes vector values. Specifically, the paper considers random feature ridge regression and derive a general bound for the population risk functional, based on which the paper gives several results such as convergence rates for well-specified and mis-specified problems, statistical consistency as well as almost sure bounds. The main results show that $O(\sqrt{N})$ random features are sufficient to derive error bounds of the order $O(1/\sqrt{N})$, where $N$ is the number of training examples. This extends the existing results from real-valued output to vector-valued outputs.


**Strengths:**

The paper provides comprehensive analysis and consider approximation, generalization, misspecification and noisy observations. Furthermore, the paper implies minimax optimal convergence rates in the well-specified setting.

The paper is very well written, and the analysis seems to be rigorous.

The technique does not rely on the explicit RF-RR solution formula, and avoids the use of matrix concentration inequalities.

**Weaknesses:**

While the technique does not use matrix concentration inequalities, the paper only considers random feature ridge regression with square loss. Therefore, the problem considered in the paper is a bit limited. It is not quite clear to me whether the technique developed here can be extended to learning with other loss functions, e.g., logistic loss. Can we get similar error bounds for vector-valued learning with random features and logistic loss?

In Theorem 3.12, the error bounds are of the order $\lambda^{r\land 1}$. Therefore, the error bounds would improve if we have very nice regularity with $r>1$. In other words, the results suffer from a saturation phenomenon. It is not clear to me whether the technique developed here can overcome this saturation phenomenon.

The obtained bounds do not show an explicit dependency on the output dimension $p$. It is not clear to me how the output dimension would affect the performance of learning with vector-valued outputs.

Typo:

Above Eq (3.5): "optimal the sense" should be "optimal in the sense"

Above Eq (3.5): "a short calculation deliver" should be "a short calculation delivers"

**Questions:**

Can the analysis here be extended to learning with logistic loss functions. In this case, we also do not have explicit formula for the solution?

Can we derive improved results if $r>1$ to overcome the saturation phenomenon?

How would the output dimension affect the convergence rates of the learning with vector-valued outputs.


**Limitations:**

I do not see concerns on negative societal impact.

---

> ### Author Rebuttal · Authors · 2023-08-08
>
>
> We start by thanking the reviewer for your appreciation of the merits of our paper and your welcome suggestions to improve it. We will correct the identified typos in the revision. Below, we address the concerns raised by the reviewer and thank the reviewer in advance for their patience in reading our detailed reply.
> 1. Closely related analysis can indeed be used to derive bounds for other settings, in particular this has been done for Lipschitz continuous loss functions; relevant results have e.g. been obtained in [SGT18]. However, the available rates for a general Lipschitz loss are significantly worse than the rates we have achieved through our refined analysis in the quadratic (i.e., squared $L^2$ loss) setting. It is unclear to what extent our improved rates may be specific to the quadratic setting. One might conjecture that similarly improved rates (as compared to the general Lipschitz case), could at least be possible when replacing the quadratic loss by e.g. a quartic, or more general $L^q$-loss, instead. We plan to extend our analysis in this direction, in the future.
> 2. We thank the reviewer for their comments regarding the saturation phenomenon. We believe that this saturation is inherent to the choice of regularizer in ridge regression, and not a mere artifact of our analysis. We would like to expand on the reason for this intuition in more detail: if we neglect errors due to a finite number of data pairs and random features (i.e. formally take the limit $N,M\to \infty$), we arrive at the ridge regression problem (B.6) on the RKHS with $\vartheta := \lambda$. This problem is explicitly solvable, with solution ${\mathcal{G}\_{\vartheta}}$ given in (B.4). Following the argument of Lemma B.3, the corresponding $L_{\nu}^2$ squared approximation error $\Vert \mathcal{G} - \mathcal{G}_{\vartheta}\Vert^2$ is generally of order $\vartheta \sum_j \lambda_j^r/(\lambda_j + \vartheta) \ge \vartheta \lambda_1^r/(\lambda_1 + \vartheta)$. For small $\vartheta$, say $\vartheta < \lambda_1$, this gives a lower bound on $\Vert \mathcal{G} - \mathcal{G}_{\vartheta}\Vert^2\ge C\vartheta$ with $C = C(\lambda_1,r)>0$, which decays at best linearly in $\vartheta$, even if $r>1$. Replacing $\vartheta$ by $\lambda$ thus strongly suggests that our upper bound of the form $\lambda^{r\wedge 1}$ for the RFM should be accompanied by a (matching) lower bound of order $\lambda$ for $r>1$.
>
> 3. Generically, the output dimension does not affect the convergence rates in the vector-valued setting, only the constants in the bounds (e.g, through the norm on the space $\mathcal{Y}$). Indeed, when $\mathcal{Y}$ is even infinite-dimensional, we show in the numerical experiments (attached figure pdf file) that the error is not sensitive to discretized output dimension $p\gg 1$. However, there exists other problems where the constants may blow up as $p\to\infty$, which would lead to a vacuous error bound. The underlying mathematical objects should have a well-defined meaning in this limit (e.g., taking the discretization resolution to infinity) to prevent this from happening.
>
> In general, as alluded to in the conclusion of our submitted manuscript, we agree with the reviewer that determining conditions under which faster rates can be achieved is a very interesting and practically relevant question, which we plan to follow up on in future work. In particular, we believe faster rates can be achieved in our analysis, under certain conditions on the probability measure on random features, and utilizing the concept of local Rademacher complexity; we would like to mention that the results of our paper could have been obtained by invoking global Rademacher complexity estimates (this is implicit in certain steps of our proofs).
>
> We sincerely hope that we have addressed the concerns of the reviewer to your satisfaction.
>
> * [SGT18] Sun, Y., et al., "But how does it work in theory? Linear SVM with random features", *Advances in Neural Information Processing Systems*, **31**, (2018)

---

> > ### Comment · Reviewer_HQhU · 2023-08-19
> > **Thx for the response**
> >
> > Thank you for providing the point-to-point response, which are satisfactory to me.

---

> > > ### Author Response · Authors · 2023-08-19
> > >
> > > We thank the reviewer for your reply. We are at your disposal during the discussion period if you have any further questions.

---

### Official Review · Reviewer_efr3 · 2023-06-26

**Soundness:** 4 excellent
**Presentation:** 4 excellent
**Contribution:** 4 excellent
**Rating:** 9
**Confidence:** 3

**Summary:**

This paper presents a theoretical error analysis of infinite-dimensional input-output ridge regression with vector-valued random features. This one is the first analysis adapted to infinite-dimensional outputs and even improves existing results for finite-dimensional outputs. Several by-products come with the main error bound, such as strong consistency in the misspecified setting and minimax optimal convergence rate in the well-specified setting. In addition, in the well-specified setting, the proposed analysis provides the sharpest parameter complexity, that is square root of the sample size. The analysis is based on a new proof technique, which is sketched in the last section.

**Strengths:**

The subject addressed by the authors is topical since there is a lack of theoretical analysis of random Fourier methods with infinite-dimensional output in the literature, and that the random Fourier technique is definitely the most practical way to speed up kernel methods.
The contribution of this work is substantial since, besides providing the first result for infinite-dimensional outputs, it also presents an improved parameter complexity ($\sqrt N$, where $N$ is the sample size) with respect to the literature ($\sqrt N \log N$).
Moreover, this paper is very well written: there is no typo and a clear effort has been made to present clearly the theoretical analysis. Remarks and examples help understanding the results and generally anticipate the questions that may arise while reading the manuscript.

**Weaknesses:**

Except for a minor remark (RF-RR Lines 48, 74, 81 is used before being defined Line 112), I think that the paper has no flaw. I must admit that I am used to seeing a numerical experiment confirming the convergence rate obtained theoretically, but I understand that choices have to be made to fit the page limit and I am quite grateful to the authors to try to explain the successive steps of their proof instead. In a way, the paper is more consistent this way.

**Questions:**

1) Could the authors give some details regarding the relaxation of the independence assumption in light of the proof outline (Section 4)?
2) Why the notation $\Phi(u; \hat \alpha, \{\theta_m\})$ is preferred to $\Phi(u; \hat \alpha)$ before Example 3.9, and the converse after? Besides, why Theorem 3.7 is named as such rather than as a corollary?
3) Could the authors explain the assumption $\mathcal G \in \operatorname{Im}(\mathcal K^{r/2})$ and its implications?

**Limitations:**

Overall, technical assumptions are discussed. Nevertheless, Societal impact is not addressed.

---

> ### Author Rebuttal · Authors · 2023-08-08
>
> We start by thanking the reviewer for your appreciation of the merits of our paper and your welcome suggestions to improve it. We will fix the use of RF-RR acronym in a revised version of the paper, thank you for catching that. Additionally, based on suggestions from the other reviewers, we have included some numerical experiments to visualize the convergence behavior of vector-valued RF-RR and compare it to our theory (see the attached figures pdf file).
> Below, we address the specific concerns raised by the reviewer and thank the reviewer in advance for their patience in reading our detailed reply.
> 1. Regarding the independence assumption, we thank the reviewer for directing our attention back to this point; in fact, thanks to the reviewer's suggestion, we have now realized that our derivation goes through almost verbatim, with only two minor modifications to the proofs in Appendix D.1, when only assuming that the data pairs $(u,y)$ are of the form $y = \mathcal{G}(u) + \eta$, where the noise $\eta$ is a subexponential random variable such that the condition expectation $\mathbb{E}[\eta|u] = 0$. This allows us to completely eliminate the independence assumption in our revised version. Thank you!
> 2. We define the equivalence of the two notations in line 104. However, we agree with the reviewer that our use is not consistent. The notation $\Phi(u;\alpha,(\theta_m))$ is introduced to emphasize the dependence of the model on the realizations $\(\theta_m)_m\sim\mu^{\otimes M}$, and Thm. 3.4 explicitly mentions the $(\theta_m)$ in its hypotheses. However, the remaining results do not explicitly mention $(\theta_m)$ in the theorem statements so we will consistently use $\Phi(u;\alpha)$ after Thm. 3.4 in the revision.
> Also, we choose to name Thm. 3.7 as is instead of Corollary 3.7 because it is the highlight/takeaway result of the paper that is easiest to compare to prior work (as in Table 1) and explain to general audiences.
> 3. Regarding the assumption $\mathcal{G} \in \mathrm{Im}(\mathcal{K}^{r/2})$, we view it as a regularity assumption on the underlying operator. In specific settings, when approximating an underlying function in finite dimensions, it corresponds to a fractional (Sobolev) regularity of the underlying function. In general, it appears difficult to check this condition, which is a definite limitation of this result, and we will point this out more explicitly in our revised version.
>
> We sincerely hope that we have addressed the concerns of the reviewer to your satisfaction.

---

> > ### Comment · Reviewer_efr3 · 2023-08-13
> >
> > I would like to thank the authors for answering my few questions and for going beyond by adding a numerical experiment supporting the theory.

---

> > > ### Author Response · Authors · 2023-08-15
> > >
> > > We thank the reviewer for your reply and we are at your disposal during the discussion period if you have any further questions.

---

### Official Review · Reviewer_1Snc · 2023-07-06

**Soundness:** 3 good
**Presentation:** 2 fair
**Contribution:** 3 good
**Rating:** 6
**Confidence:** 4

**Summary:**

This paper investigates the theoretical aspects of learning vector-valued operators with random features. Specifically, it studies the convergence of the random feature estimate $\Phi(u;\hat{\alpha})$ to the true underlying function $\mathcal{G}$. Its main results are:

Theorem 3.4: Error bound for fixed $\lambda$ and fixed $N$.

Theorem 3.7: The same error bound in the well-specified case $\mathcal{G}\in\mathcal{H}$.

Theorem 3.10: Almost sure consistency of the random feature estimate to the true function $\mathcal{G}$ as $N\rightarrow\infty$ and $\lambda\rightarrow0$.

Theorem 3.12: The convergence rate.

**Strengths:**

The paper is entirely a theoretical paper, so the value of the paper should lie in the significance of the results and the correctness of its proofs. I believe that the stated results would be good contributions (although see Questions for Theorem 3.10), but perhaps quite limited in scope. Unfortunately, I have not had time to go through all of the proofs in detail (perhaps I will in the subsequent reviewing process).

**Weaknesses:**

The paper is in general well-written, and the mathematics is presented quite clearly. However, its structure makes it rather difficult to read. For example, when reading the main body of the paper, we are led to the appendix several times for definitions that are required to read on in the main body. This happens on L151, L177, L179, L222 and L240. Having proofs in the appendix is fine, and is often done, but the way we are led to the appendix in this paper disrupts the flow a bit in my opinion. Also, the proofs are not in one place - we have to jump back and forth between Section 4 and the Appendix to read the proof of the results.

Also, the results are very abstract, and I think it would have been much better if some concrete examples of $\phi$, $\Phi$, $\theta$, $\alpha$ and $\mathcal{H}$ were given. An example of a learning algorithm to which the results could have been applied would have been even better.

Minor comments:

L228: I think the superscript $\lambda$ should be $\lambda_k$.

L614, displayed equation: The squared brackets after $\mathbb{E}$ are not closed.

**Questions:**

Theorem 3.10 and Corollary 3.13: It is, at least in my limited experience, rare to see almost sure statements in statistical learning theory. Could the authors please comment on what made it possible in this case, or whether they believe a simple Borel-Cantelli argument should make all of the results in statistical learning theory almost sure statements?

Also, none of the results seem to depend on the distribution $\mu$ of $\theta$, and always just the number $M$ of them. I could be mistaken, but this is hard to believe - what if $\mu$ put all mass on one point? I didn't seem to be able to find in Assumptions 3.1, 3.2 and 3.3 that prevented something like this.

**Limitations:**

The authors barely discuss the limitations of their work, except as "future work" in the Conclusion section.

---

> ### Author Rebuttal · Authors · 2023-08-08
>
> We start by thanking the reviewer for reading our paper and for their comments. We have fixed the typos in the revised version of our paper. Regarding the reviewer's criticism of the structure and readability of the paper, we agree that splitting into proofs in appendix and theorem/lemma statements in the main body make a thorough reading difficult. To ease the burden on the reader, in the revision we will: 1) use the ``thm-restate'' LaTeX package that will repeat the full theorem/lemma statements immediately above their proofs in the appendix, so no more need to flip back and forth; and 2) Submit the full paper (main body plus supplement) as the official supplementary materials so that all hyperlinks work as intended when reading results, proofs, and definitions.
>
> Additionally, we have now included a concrete operator learning benchmark example (Burgers' evolution operator) where our RF ridge regression algorithm is actually implemented. Here, $p=\infty$ and a nontrivial choice of $\varphi$, $\theta$, and $\Phi$ are used in the regression problem (see the attached pdf figure file for the details).
>
> Below, we address the specific concerns raised by the reviewer and thank the reviewer in advance for their patience in reading our detailed reply.
> 1. Regarding the reviewer's question on Borell-Cantelli, we believe that it is indeed the case that non-asymptotic, high-probability results can often be turned into almost sure asymptotic results. However, there are certain draw-backs to such almost-sure asymptotic results. For example, even though the asymptotic behavior is guaranteed to occur ``eventually'' (with probability 1), it is completely unclear whether we should ever expect to observe such rates at practically relevant numbers of data-pairs and random features. In this sense, we view our asymptotic consistency result as a considerably weaker assertion than the high-probability estimates that it is derived from.
> 2. Regarding the reviewer's comment about the dependence on the underlying measure $\mu$, it is indeed true that our results do not make any specific assumptions on $\mu$, as such. However, $\mu$ implicitly determines the underlying reproducing kernel Hilbert space (RKHS), via the random feature map $\varphi$. In particular, if $\mu$ is concentrated on a single point, then this RKHS is necessarily at most one-dimensional, and hence our results only imply convergence of the RFM to any operator *belonging to this one-dimensional subspace*, in this specific case. More generally, our results require either that the underlying operator belong to the RKHS or that it at least belong to the $L^2_\nu$-closure of this RKHS. This can be viewed as a compatibility requirement between the operator and the RFM; the performance of the RFM should absolutely be expected to depend on the "degree of compatibility between the operator $\mathcal{G}$, the measure $\mu$ and the random feature map $\varphi$". We thank the reviewer for this pertinent question, and will include a more detailed discussion of this point in our revision.
>
> We sincerely hope that we have addressed all your concerns and kindly request the reviewer to update their assessment accordingly.

---

> > ### Comment · Reviewer_1Snc · 2023-08-13
> > **Thank you**
> >
> > Thank you for your answers to my questions, and the effort to demonstrate a concrete example in the Figure provided. I am quite satisfied with the answers to my questions, and I have raised my score accordingly.

---

> > > ### Author Response · Authors · 2023-08-15
> > >
> > > We thank the reviewer for their comments as well as for increasing our score.

---

### Official Review · Reviewer_ykP5 · 2023-07-12

**Soundness:** 4 excellent
**Presentation:** 4 excellent
**Contribution:** 4 excellent
**Rating:** 8
**Confidence:** 4

**Summary:**

The paper proposes a statistical analysis of the risk associated to learning with vector-valued random features (vv-RF) in the context of ridge regression.

The analysis shows that $\sqrt{N}$ random features are enough to attain a $\mathcal{O}(\frac{1}{\sqrt{N}})$ squared error (matching minimax rates) and strong consistency in the well-specified setting, as well as slower convergence rates related to how well the target function can be approximated in the RKHS in case of misspecified setting.

**Strengths:**

- Topic is highly relevant to the machine learning community
- Paper is remarkably well written despite the technical complexity
- The paper is mathematically sound
- Assumptions are mild and very standard

**Weaknesses:**

Contribution (C4) is a bit exaggerated in my opinion, see questions.

**Questions:**

In the contributions listing, you mention in (C4) that the proof approach is not specific to RR. While I agree that you make no use of the closed form solution, you still make use of the fact that the loss is the squared loss (e.g. in 4.4), and the analysis would not hold if another loss was chosen. Same remark applies in the conclusion when you mention that future works could focus on general $L^p$ losses. If the analysis is not specific to the RR setting, for which class of losses does it hold ?

And more of a remark: it would be good to include a non trivial example of a feature map $\phi(x, \theta)$ used to solve a problem where $p = \infty$.

**Limitations:**

- Limitations are well discussed in the paper, expect perhaps the point about the specificity to the RR that is mentioned in the "Questions" section.

---

> ### Author Rebuttal · Authors · 2023-08-08
>
> We start by thanking the reviewer for your appreciation of the merits of our paper and your welcome suggestions to improve it. Below, we address the concerns raised by the reviewer and thank the reviewer in advance for their patience in reading our detailed reply.
> 1. We believe the reviewer's comment about contribution (C.4) is a very fair criticism, as certain steps in our derivation, including identity (4.4), are indeed specific to a quadratic loss. In view of this, we will tone down the statement of (C.4) in a revised version of the paper. We plan to expand on error bounds for the RFM for more general loss functions in future work. At least in the absence of noise, our intuition is that the RFM estimates should exhibit the same scaling as estimates for Monte-Carlo sampling; based on this intuition, we fully expect generalization to $L^q$-type loss functions to be possible since the Monte-Carlo error is independent of $q$. When allowing for noise, significant adaptations of our arguments may indeed be necessary, as equation (4.4), which is specific to a quadratic loss, is then important to our present approach.
> 2. A non-trivial example of a feature map in the $p=\infty$ infinite-dimensional output space case is given by the Fourier space features [NS21, Eqn. 3.5] for the Burgers' equation solution operator. It leads to a fully correlated (non-diagonal) limiting operator-valued kernel (hence non-trivial), which is important for the setting $p=\infty$. We include numerical results for this problem in the attached pdf figure file, and will add this concrete example to the revised version of our paper.
>
> We again thank the reviewer for appreciating the merits of this work, and sincerely hope to have addressed your concerns.
>
> * [NS21] Nelsen, N.H., Stuart, A.M., "The random feature model for input-output maps between Banach spaces", *SIAM Journal on Scientific Computing*, **43**(5), A3212--A2343, (2021)

---

> > ### Comment · Reviewer_ykP5 · 2023-08-19
> > **Acknowledging rebuttal**
> >
> > I thank the authors for their answer. My original assessment still holds, I advocate for acceptance.

---

> > > ### Author Response · Authors · 2023-08-19
> > >
> > > We thank the reviewer for their reply, and for appreciating the merits of our work. We remain at your disposal during the discussion period if you have any further questions.

---

### Official Review · Reviewer_1g4u · 2023-07-25

**Soundness:** 3 good
**Presentation:** 3 good
**Contribution:** 3 good
**Rating:** 6
**Confidence:** 3

**Summary:**

The paper proposes a learning theory for ridge regression in random feature models in the setting when the input-output map $\mathcal{G}: \mathcal{X} \mapsto \mathcal{Y}$ is vector-valued (potentially infinite-dimensional). The model under consideration is a random feature model $\phi: \mathcal{X} \times \Theta \mapsto \mathcal{Y}$, such that the feature map $\Phi(\cdot)$ consists of a linear combination of RFs $\Phi(\cdot) = \sum_{m=1}^M \alpha_m \phi(\cdot, \theta_m)$,  with different random weights sampled from $\theta_1, \dots, \theta_M \sim \mu$. The parameters $\alpha_1, \dots, \alpha_M$ are fitted by minimizing the regularized empirical risk $\frac{1}{N} \sum_{n=1}^N \lVert y_n - \Phi(u_n) \rVert_{\mathcal{Y}}^2 + \frac{\lambda}{M} \lVert{\alpha}\rVert_2^2$ given data sampled from some data generating distribution $(x_i, y_i) \sim \nu$. The assumed hypothesis class is an RKHS that consists of functions $\mathcal{F} \in \mathcal{H}$ given by weighted averages of the RFs $\mathbb{E}_{\mathcal{P}(\Theta)}\lbrack \alpha(\theta) \phi(\cdot, \theta)\rbrack$ for $\alpha(\cdot) \in L^2_\mu(\Theta, \mathbb{R})$. However, the considered setting also allows for model misspecification $\mathcal{G} = \mathcal{G}_\mathcal{H} + \rho$, such that $\mathcal{G}_\mathcal{H} \in \mathcal{H}$ and $\rho$ is almost surely bounded. The observation noise $y = \mathcal{G}(u) + \eta$ is modelled by a subexponential distribution allowing for heavier noise than the subGaussian.

Given the previous setting, the main result seems to be Theorem 3.4, which bounds the population squared error $\mathbb{E}_\nu\lbrack \mathcal{G}(u) - \Phi(u)\rbrack$ in terms of various quantities relating to the functions $\mathcal{G}, \mathcal{G}_\mathcal{H}, \rho$, the observation noise $\eta$ under mild conditions which roughly state that $M$ is of order $\sqrt{N}$. In contrast to previous work, the parameter complexity $M$ is free from logarithmic factors, which gives the lowest parameter complexity so far. As an application of the previous bound, the authors results regarding 1) strong statistical consistency of the model, that is, almost sure convergence in mean square of $\Phi(\cdot)$ to $\mathcal{G}(\cdot)$, and 2) corresponding explicit convergence rates. Sketches of the proofs are given in Section 4 to guide the reader along the main steps.

**Strengths:**

The paper is well-written and coherent. The area, theoretical properties of infinite-dimensional RF models, is relevant to (contemporary) applications in scientific computing, such as learning the solution operator of PDEs using RF or neural models. The presented results are highly technical and non-trivial to derive. In addition to generalizing the theory of RF models to this setting, the RF parameter complexity is also sharper by a logarithmic factor as opposed to previous works, thatworked with stronger assumptions on the output domain. The presentation is good-ish, I think they did a good job in outlining the setting, then the main results, and presenting the key steps of the proofs in section with full proofs deferred to the appendix.

**Weaknesses:**

As a theoretical work, there is no empirical part to speak of. This, on the one hand is common in these types of papers, but on the other, it makes the paper less accessible to a majority of the NeurIPS community. Although it seems like such RF models do have important applications, I am unsure about the size of the community that is involved in these kinds of niche results, as the main application only seems to be in operator learning.

One other point is that although there are applications listed, the mathematical objects considered in the paper still seem kind of abstract, I think giving more examples and connecting the dots more to the applications would help in this aspect (see question 1).

Lastly, no illustrations are provided whatsoever to support comprehension or visualize the results. Some figures would go a long way to make the paper less dense (see question 2)

**Questions:**

- As being familiar with random feature models, I understood the main premise of the paper, but I was still left wondering about how one could construct feature maps into such infinite-dimensional spaces. I think giving some examples about what kind of RFs were used before for operator learning, on what kind of data, etc would go a long way.

- I was wondering if it would be possible to visualize the various hyperparameter complexities and empirically validate the convergence rate appearing in the theorems to have some visual intuition either using synthetic data or one of the datasets appearing in previous work (if the true operator is known)? Another idea which would also help in structuring the results and aid in understanding them is to draw a flowchart about the previous theorems and lemmas showing how they depend on each other.

Minor:
- In line 32, when the authors state the size of RF matrices is quadratic in the number of features $M$, do they actually mean the covariance matrix that has size $M \times M$, or the $N \times M$ feature matrix? In the latter case I am unsure where the quadratic factors comes in.
- In line 228, there might be a $k$ missing in $\mathfrak{R}_N^{\lambda_k}$, otherwise $\hat \alpha$ does not actually depend on $k$
- In Apprendix A, line 489, the authors define the subexponential norm in terms of the moments, which is slightly uncommon, since it's standard to define it in terms of the exponential Orlicz norm. It might be beneficial to state the original definition first, and relate that

**Limitations:**

As a purely theoretical work, there is no direct societal impact commonly associated with AI models. The authors have addresed the limitations.

---

> ### Author Rebuttal · Authors · 2023-08-08
>
> We start by thanking the reviewer for your appreciation of the merits of our paper and your welcome suggestions to improve it. Below, we address the concerns raised by the reviewer and thank the reviewer in advance for their patience in reading our detailed reply.
> 1. The construction of "good" random feature pairs $(\varphi,\mu)$ in the infinite-dimensional output space setting is still an open problem. This is partly because there are no ``canonical'' operator-valued kernels (OVKs) $K\colon\mathcal{X}\times\mathcal{X}\to\mathcal{L}(\mathcal{Y};\mathcal{Y})$, which the random features approximate. In contrast, canonical kernels in the scalar output setting include squared exponential and Mat\'ern kernels, which have interpretable hyperparameters such as lengthscales, regularity, and variances that may be adapted to the problem. In the operator learning setting, handcrafted vector-valued random features were designed in [NS21, Sec. 3] for specific PDE problems on a case-by-case basis. In [KDPCRA16], separable OVKs $K(u,u')=k(u,u')T$ were used, where $k$ is a canonical scalar kernel (hence RF approximations of $K$ are then straightforward if one for $k$ is known) and $T$ is a bounded linear operator on $\mathcal{Y}$. Both papers work with functional data. More general constructions would require novel ways to adapt such features or kernels to data.
> 2. To empirically test the validity and sharpness of our theory, we take the reviewer's suggestion and implement a Burgers' equation operator learning benchmark for a range of sample sizes $N$, random features $M$, and data resolutions $p$. The dataset appears in previous work on operator learning. This example does not necessarily satisfy our theoretical assumptions because we cannot verify that the Burgers' solution operator $\mathcal{G}$ is an element of the RF's RKHS. Also, the feature map uses an unbounded activation function (ELU), while our theory is only developed for bounded RFs. Nevertheless, the observed parameter and sample complexity reasonably validate the theory.
> 3. Regarding the reviewer's suggestion of including a flowchart illustrating the interdependency of our results, we agree that this would greatly help in providing a quick overview of our theoretical results. We thank the reviewer for this suggestion, and will include it in a revised version of our paper.
>
> Regarding the minor comments:
> * By quadratic, we do mean the RF ``gram/covariance-like'' matrices $k_{ij}=\frac{1}{N}\sum_{n=1}^N\langle \varphi(u_n;\theta_i),\varphi(u_n;\theta_j)\rangle_{\mathcal{Y}}$ that are of size $M$ by $M$.
> * We thank the reviewer for pointing out the typo in Line 228.
> * In the revision, we will cite the standard definition of exponential Orlicz norm that the reviewer mentions (e.g., Vershynin's book).
>
> We sincerely hope to have addressed the concerns to your satisfaction and thank the reviewer again for pointing them out to us.
>
> * [NS21] Nelsen, N.H., Stuart, A.M., "The random feature model for input-output maps between Banach spaces", *SIAM Journal on Scientific Computing*, **43**(5), A3212--A2343, (2021)
> * [KDPCRA16] Hachem, K., et al., "Operator-valued kernels for learning from functional response data", *Journal of Machine Learning Research*, **17**, 1-54, (2016)

---

### Author Rebuttal · Authors · 2023-08-08

At the outset, we would like to thank all five reviewers for their thorough and patient reading of our article. Their fair criticism and constructive suggestions will enable us to improve the quality of our article. If accepted, a revised camera-ready version of the article, with changes as outlined below, will be uploaded. We proceed to answer the points raised by each of the reviewers individually, below. We also attach one page of figures that show the results of numerical experiments suggested by the reviewers (with code to be made publicly available on Github). The literature references in this figures document refer to the two papers below.

Yours sincerely,

Authors of "Error Bounds for Learning with Vector-Valued Random Features".

* [KLLABSA23] Kovachki, N., Li, Z., et al., "Neural operator: Learning maps between function spaces with applications to PDEs", *Journal of Machine Learning Research*, **24**(89), 1--97, (2023)
* [NS21] Nelsen, N.H., Stuart, A.M., "The random feature model for input-output maps between Banach spaces", *SIAM Journal on Scientific Computing*, **43**(5), A3212--A2343, (2021)

---

### Decision · Program_Chairs · 2023-09-21

**Decision:**

Accept (spotlight)

**Comment:**

The submission is about scaling up kernel methods. The authors consider the random feature (RF) approximation of kernel ridge regression with Polish input space and separable Hilbert output space. They prove a high probability error bound on the population squared error (Theorem 3.4). This result implies (i) O(1/sqrt{N}) convergence rate of the squared error in the well-specified case (Theorem 3.7) with O(\sqrt{N}) RFs where N denotes the sample size, (ii) strong consistency in the misspecified case (Theorem 3.10), and (iii) slower rates in the misspecified setting (Theorem 3.12).

Kernel methods are among the most powerful tools of machine learning; scaling them up in a principled way is of central importance to the community. The established novel tight results in the submission nicely contribute to this endeavor.